# Loss of zebrafish *atp6v1e1b*, encoding a subunit of vacuolar ATPase, recapitulates human ARCL type 2C syndrome and identifies multiple pathobiological signatures

**Lore Pottie**[1,2], **Wouter Van Gool**[1,2], **Michiel Vanhooydonck**[1,2], **Franz-Georg Hanisch**[3], **Geert Goeminne**[4,5], **Andreja Rajkovic**[6], **Paul Coucke**[1,2], **Patrick Sips**[1,2ᵒ], **Bert Callewaert**[1,2ᵒ] *

**1** Center for Medical Genetics Ghent, Ghent University Hospital, Ghent, Belgium, **2** Department of Biomolecular Medicine, Ghent University, Ghent, Belgium, **3** Institute of Biochemistry II, Medical Faculty, University of Cologne, Cologne, Germany, **4** VIB Metabolomics Core Ghent, Ghent, Belgium, **5** Department of Plant Biotechnology and Bioinformatics, Ghent University, Ghent, Belgium, **6** Department of Food technology, Safety and Health, Faculty of Bioscience Engineering, University of Ghent, Ghent, Belgium

ᵒ These authors contributed equally to this work.
* Bert.Callewaert@Ugent.be

**Data Availability Statement:** RNA sequencing data used for the gene expression analysis of both

## Abstract

The inability to maintain a strictly regulated endo(lyso)somal acidic pH through the proton-pumping action of the vacuolar-ATPases (v-ATPases) has been associated with various human diseases including heritable connective tissue disorders. Autosomal recessive (AR) cutis laxa (CL) type 2C syndrome is associated with genetic defects in the *ATP6V1E1* gene and is characterized by skin wrinkles or loose redundant skin folds with pleiotropic systemic manifestations. The underlying pathological mechanisms leading to the clinical presentations remain largely unknown. Here, we show that loss of *atp6v1e1b* in zebrafish leads to early mortality, associated with craniofacial dysmorphisms, vascular anomalies, cardiac dysfunction, N-glycosylation defects, hypotonia, and epidermal structural defects. These features are reminiscent of the phenotypic manifestations in ARCL type 2C patients. Our data demonstrates that loss of *atp6v1e1b* alters endo(lyso)somal protein levels, and interferes with non-canonical v-ATPase pathways *in vivo*. In order to gain further insights into the processes affected by loss of *atp6v1e1b*, we performed an untargeted analysis of the transcriptome, metabolome, and lipidome in early *atp6v1e1b*-deficient larvae. We report multiple affected pathways including but not limited to oxidative phosphorylation, sphingolipid, fatty acid, and energy metabolism together with profound defects on mitochondrial respiration. Taken together, our results identify complex pathobiological effects due to loss of *atp6v1e1b in vivo*.

## Author summary

Cutis laxa syndromes are pleiotropic disorders of the connective tissue, characterized by skin redundancy and variable systemic manifestations. Cutis laxa syndromes are caused

atp6v1e1b-deficient zebrafish and WT controls have been deposited in the ArrayExpress database at EMBL-EBI under accession number E-MTAB-8824, and can be accessed at the following link: https://www.ebi.ac.uk/arrayexpress/experiments/E-MTAB-8824.

**Funding:** B.C. is a senior clinical investigator of the Research Foundation Flanders. This work was supported by a starting grant of the Special Research Fund from Ghent University (grant 01N04516C to B.C.), by a Methusalem Grant (BOFMET2015000401) from Ghent University, by a junior fundamental research project grant of the Fund for Scientific Research (G035620N) and by the European Union's Horizon 2020 research and innovation program, under the Marie Skłodowska-Curie grant agreement No. 794365 to P.S. The funders had no role in study design, data collection and analysis, decision to publish, or preparation of the manuscript.

**Competing interests:** The authors have declared that no competing interests exist.

by pathogenic variants in genes encoding structural and regulatory components of the extracellular matrix or in genes encoding components of cellular trafficking, metabolism, and mitochondrial function. Pathogenic variants in genes coding for vacuolar-ATPases, a multisubunit complex responsible for the acidification of multiple intracellular vesicles, cause type 2 cutis laxa syndromes, a group of cutis laxa subtypes further characterized by neurological, skeletal, and rarely cardiopulmonary manifestations. To investigate the pathomechanisms of vacuolar-ATPase dysfunction, we generated zebrafish models that lack a crucial subunit of the vacuolar-ATPases. The mutant zebrafish models show morphological and functional features reminiscent of the phenotypic manifestations in cutis laxa patients carrying pathogenic variants in *ATP6V1E1*. In-depth analysis at multiple -omic levels identified biological signatures that indicate impairment of signaling pathways, lipid metabolism, and mitochondrial respiration. We anticipate that these data will contribute to a better understanding of the pathogenesis of cutis laxa syndromes and other disorders involving defective v-ATPase function, which may eventually improve patient treatment and management.

## Introduction

The intraluminal pH of specific subcellular compartments is maintained by the proton-pumping action of the vacuolar-ATPases (v-ATPases), thereby regulating a range of molecular processes including activation of enzyme activity, protein folding, vesicle trafficking, and support of organelle function and integrity. V-ATPases consist of ubiquitously expressed multisubunit complexes, composed of a membrane-associated $V_0$ domain and a cytosolic (catalytic) $V_1$ domain. Hydrolysis of ATP by the complex generates proton pumping activity against a concentration gradient in the lumen of endo(lyso)somes, secretory vesicles, Golgi apparatus and across the plasma membrane into the extracellular space surrounding specialized cells including osteoclasts, epididymal clear cells and renal epithelial intercalating cells [1].

Various human diseases are linked to defects in genes encoding subunits of the v-ATPase, including connective tissue disorders (e.g. autosomal recessive (AR) cutis laxa (CL) syndrome type 2A (MIM: 219200) (*ATP6V0A2* [MIM: 611716]), ARCL type 2C (MIM:617402) (*ATP6V1E1* [MIM: 108746]), and ARCL type 2D (MIM:617403) (*ATP6V1A* [MIM: 607027])) [2–12]. ARCL type 2A, 2C, and 2D show variable glycosylation abnormalities [10,13,14], hypotonia, delayed neuromotor development, variable intellectual disability, sensorineural hearing loss, skeletal abnormalities, facial dysmorphology, and cardiopulmonary involvement including pneumothorax, hypertrophic cardiomyopathy, and aortic root dilation [10,11,15,16]. The pathological mechanism underlying ARCL type 2A, 2C, and 2D is currently unknown.

Intracellular v-ATPases play a role in the regulation of key physiological cell signaling pathways. V-ATPases are involved in downstream Notch signaling [17–20], Wnt signaling [21–27], activation of mechanistic target of rapamycin complex 1 (mTORC1), and AMP-activated protein kinase (AMPK) signaling [28]. Using *in vitro* experiments, it was shown that lysosomal amino acids activate mTORC1 recruitment to the lysosomal membrane due to conformational changes of the Ragulator complex. V-ATPases are tightly associated with this complex and serve as the amino acid sensor by still unknown mechanisms [29–31]. mTORC1 signaling induces anabolic events. In contrast, if energy conditions are low, liver kinase B1 will activate AMPK located in the lysosomal membrane [32], inducing catabolic events. In addition, Notch and Wnt signaling rely on acidification of early endosomes through v-ATPases. Prorenin receptor (PRR) acts as an adaptor protein between LRP6 and v-ATPases to promote Wnt-

receptor endocytosis, phosphorylation, and subsequent Wnt/β-Catenin signaling activation [33–35]. Notch receptor cleavage occurs through secretases at the plasma membrane or in acidified early endosomes. Following translocation to the nucleus, the Notch intracellular domain forms a transcription-activation complex and initiates transcription of downstream target genes [17–19]. Finally, recent studies have highlighted a novel mechanism linking v-ATPase inhibition, hypoxia-inducible factor 1 alpha (HIF-1α) signaling and iron deficiency. *In vitro*, genetic and pharmacological inhibition of v-ATPases impair endo(lyso)somal acidification which leads to defective cellular iron uptake via the transferrin system. Iron depletion results in reduced prolyl hydroxylase activity, leading to stabilization of HIF-1α and activation of HIF-1α-dependent signaling mechanisms [36–38]. However, it remains to be seen if these non-canonical functions of v-ATPase deficiency, altered iron homeostasis, and subsequent HIF-1α-mediated responses are also affected *in vivo* and contribute to the pathogenesis of disorders related to defects in v-ATPase function.

In order to study v-ATPase function *in vivo*, several mouse studies were established. However, complete loss of the murine proteolipid 16 kDa subunit of the v-ATPase $V_0$ complex (encoded by the *Atpv0c* gene) led to embryonic lethality, precluding a detailed investigation of the complete loss of v-ATPase function [39]. Since external fertilization and transparent development allow for a detailed characterization of embryonic development in zebrafish, we modelled genetic depletion of *atp6v1e1b* in zebrafish. Despite early mortality, this model shows phenotypic features mimicking ARCL type 2C with hypotonia, craniofacial abnormalities, vascular anomalies, N-glycosylation abnormalities, and an altered dermal connective tissue structure. As expected, *atp6v1e1b*-deficiency disrupts (non-)canonical function(s) of v-ATPases. Unbiased analysis of the transcriptome, metabolome, and lipidome furthermore indicates alterations in oxidative phosphorylation, fatty acid metabolism, sphingolipid, and energy metabolism in this *in vivo* model of v-ATPase deficiency.

## Results

### Genetic loss of *atp6v1e1b* in zebrafish leads to early mortality and mimics the human phenotype of *ATP6V1E1*-related CL syndrome

*ATP6V1E1*, which codes for a subunit of the v-ATPase that connects the transmembrane domain to the cytosolic domain, has two ohnologs in zebrafish, *atp6v1e1a* and *atp6v1e1b*. Whole-mount in situ hybridization (WISH) showed that these ohnologs share similar spatial and temporal expression patterns. WISH indicates localization of *atp6v1e1a* and *atp6v1e1b* in the cranial region, pronephros, and heart of the zebrafish embryo at multiple developmental stages (**S1A–S1C Fig**) [40]. To identify the manifestations of reduced organelle acidification *in vivo*, we disrupted the function of both ohnologs in zebrafish.

First, we generated a zebrafish line harboring a four base-pair (bp) deletion in exon 2 of *atp6v1e1a* (*atp6v1e1a*^+/-^), c.71delTGAG, which led to nonsense-mediated decay (NMD) due to a premature stop codon (**S1D and S1E Fig**). *Atp6v1e1b* expression remained stable in *atp6v1e1a*^-/-^ zebrafish while *atp6v1e1a* expression is significantly reduced (**S1E Fig**). *Atp6v1e1a*^-/-^ zebrafish are viable, do not show pigmentation defects, and generate a Mendelian distribution of the disrupted allele in their offspring. *Atp6v1e1a*^-/-^ zebrafish have a normal lifespan and do not develop bone morphological abnormalities (**S1F–S1I Fig**).

Next, we generated a zebrafish line harboring a two bp insertion followed by a three bp deletion in exon 5 of *atp6v1e1b* (*atp6v1e1b*^cmg78/+^), c.334insGG; c.337-340delCGG, which led to NMD due to a premature stop codon (**S2A and S2B Fig**). We further obtained a zebrafish line with a large retroviral insertion in the 5'UTR of *atp6v1e1b* (*atp6v1e1b*^hi577aTg/+^) interfering with mRNA transcription (**S2A and S2B Fig**) [41,42]. Atp6v1e1 protein levels are decreased in both

*atp6v1e1b*-deficient zebrafish while more residual protein is present in *atp6v1e1b*[hi577aTg/hi577aTg] (**S2C–S2E Fig**). We observed a very low level of atp6v1e1 protein in one of the *atp6v1e1b*[cmg78/cmg78] samples. Considering the sequence homology with the human ATP6V1E1 protein fragment which was used as an immunogen to generate the antibody, atp6v1e1a and atp6v1e1b proteins are recognized on immunoblot. Therefore, we conclude that the remaining signal in the *atp6v1e1b*[cmg78/cmg78] sample originates from the atp6v1e1a protein. We recapitulated a pigment dilution phenotype in *atp6v1e1b*-deficient zebrafish as an early manifestation. *Atp6v1e1b*-deficient zebrafish were unable to hatch and died at 3–5 days post fertilization (dpf) (**S2G Fig**). Upon manual removal of the chorion, the mutant zebrafish survived until 8 dpf (**S2F Fig**) albeit with a severe systemic phenotype.

This prompted us to further investigate the phenotype of the surviving *atp6v1e1b*-deficient larvae to assess whether dominant features in the clinical presentation of ARCL type 2C are present in mutant zebrafish. To evaluate hypotonia as a possible cause of hatching problems, we quantitatively assessed the touch-evoked escape response at 3 dpf, which was impaired in both *atp6v1e1b* mutant zebrafish lines (**S2H Fig**). Alcian blue staining showed maxillary and mandibular hypoplasia in both mutant zebrafish which could correlate to midface hypoplasia and micrognathia in humans suffering from ARCL type 2C (**Fig 1A**). In addition, reduction in the intensity of Alcian blue staining in both mutant zebrafish suggests changes in the expression of chondroitin sulfate proteoglycans. *Atp6v1e1b*-deficient zebrafish showed a misshapen and slightly retracted Meckel's cartilage. Moreover, the length of the Meckel's cartilage, palatoquadrate and ceratohyal structures was decreased (**Fig 1B**) and the angle between ceratohyal structures was increased (**Fig 1C**).

Next, we assessed the cardiac function and vascular patterning in the *atp6v1e1b*-deficient zebrafish at the embryonic stage, using the endothelial *Tg(kdrl:eGFP)* reporter line. Vascular anomalies were present in both zebrafish lines at 5 dpf (**Fig 1E**). We discovered a structural malformation of the aortic arches, the opercular artery and the ventral aorta which were smaller and more constricted with less vascular loops in *atp6v1e1b*-deficient zebrafish. The ventral aorta segment proximal to the bulbus arteriosus was significantly dilated in *atp6v1e1b*-deficient zebrafish (**Fig 1D**). Furthermore, *atp6v1e1b*-deficient zebrafish showed a significantly decreased stroke volume and cardiac output at 3 dpf, indicative of cardiomyopathy (**Fig 1F**). Posterior blood flow was significantly decreased in *atp6v1e1b*[hi577aTg/hi577aTg] but not in *atp6v1e1b*[cmg78/cmg78] (**Fig 1F and 1G**). Starting from 5 dpf, mutant zebrafish developed severe pericardial edema (**Fig 1A**), indicating severe cardiovascular dysfunction.

Transmission electron microscopy (TEM) analysis of the dermis from WT controls and *atp6v1e1b*-deficient zebrafish showed manifest defects around the epidermal basement membrane. First, *atp6v1e1b*[hi577aTg/hi577aTg] samples showed zones of epidermal detachment, larger primary dermal stroma, and a severely disorganized structure of collagenous fibrils (**Fig 2A**). No alteration in basement membrane length was observed. Second, *atp6v1e1b*[cmg78/cmg78] samples showed folded basement membrane, indicating skin redundancy (**Fig 2A**). RT-qPCR analysis confirmed collagen and elastin alterations in both mutant zebrafish lines at 4 dpf. *Col1a1a* and *col1a2* (**Fig 2B–2E**) were significantly reduced in *atp6v1e1b*-deficient zebrafish while *col1a1b* (**Fig 2C**) expression levels remained stable. In addition, downregulation of *elnb* expression (**Fig 2A**) occurred in both mutant zebrafish lines.

Humans with ATP6V1E1 pathogenic variants show abnormal glycosylation profiles and dilated Golgi apparati. We investigated the relative abundances of N-glycan sugar chains in proteins extracted from whole body lysates and assessed Golgi morphology by TEM analysis on samples of *atp6v1e1b*-deficient zebrafish at 3 dpf [10]. We confirmed that the N-glycome profiles in whole body zebrafish larval lysates were dominated by oligo-mannose glycans (M5–M9) and had minor abundancies of complex glycans [43–45]. Complex glycans were exclusively composed of bi-antennary glycans without core fucosylation and antennary structures

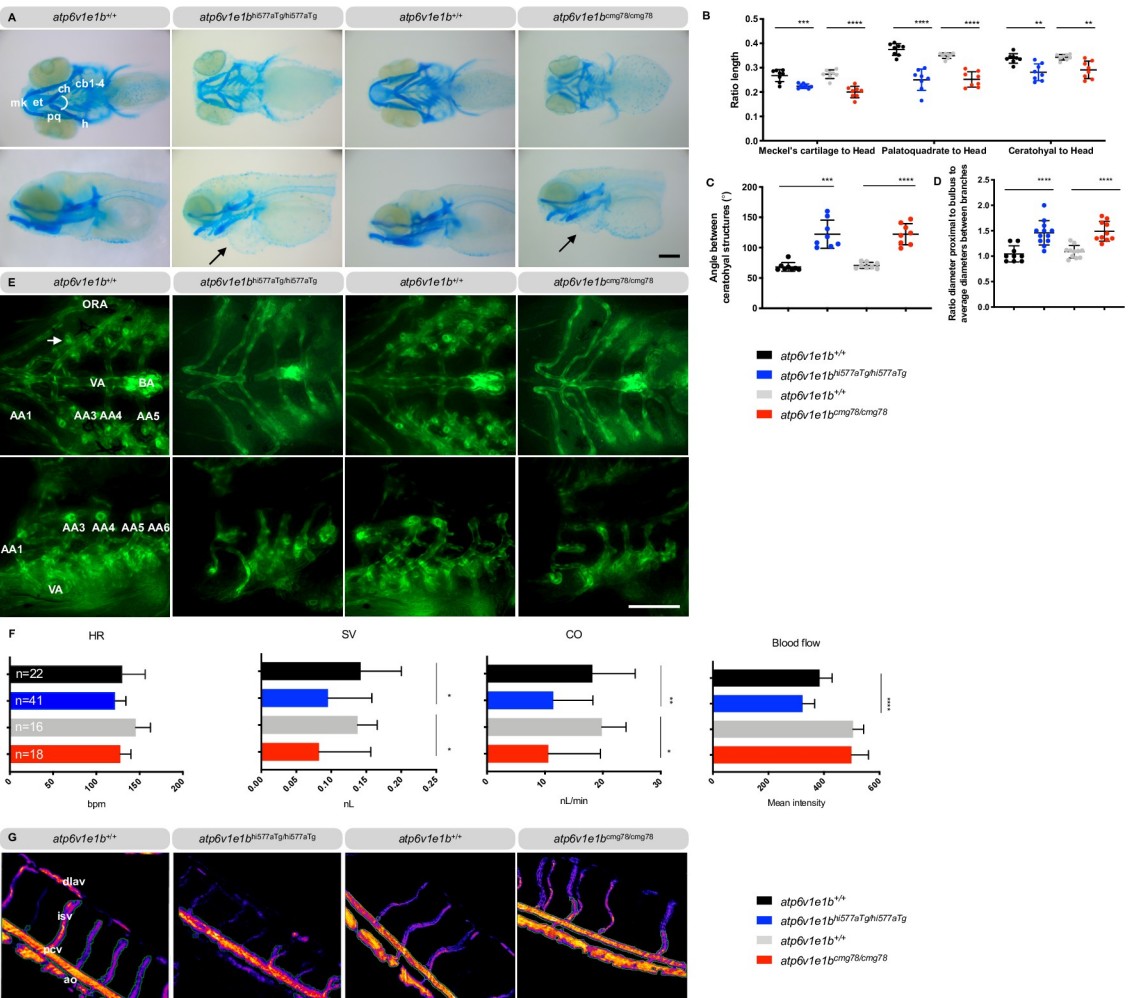

**Fig 1. *Atp6v1e1b*-deficient zebrafish have craniofacial, vascular and cardiac abnormalities reminiscent of the human *ATP6V1E1*-related CL phenotype.** Representative images are shown. (A) Ventral (top panel) and lateral views (bottom panel) of Alcian blue stained craniofacial structures at 5 dpf reveal misshapen and shorter Meckel's cartilage (m), shorter palatoquadrate (pq) and shorter ceratohyal (ch) structure and a higher angle between ch structures in *atp6v1e1b*-deficient zebrafish. Pericardial edema is indicated by an arrow. (B) The length of the individual cartilage structures was measured and normalized to the head length of the larvae. (C) Quantification of the angle between the ceratohyal structures. (D) Ratio of the aortic diameter proximal to the bulbus arteriosus (BA) to the average aortic diameter between the first (AA3) and the second branchial arch (AA4) as a measure of local dilatation. (E) Ventral (top panel) and lateral views (bottom panel) of the cardiovascular structure of *atp6v1e1b*-deficient zebrafish in the *Tg(kdrl:eGFP)* background at 5 dpf. (F) Cardiac parameter analysis based on brightfield microscopy recordings at 3 dpf. (G) Pseudo-colored processed images representing relative blood flow intensity in the trunk of 3 dpf zebrafish. Major blood vessel ROI used for quantification is marked with a green outline. Data are expressed as mean ± standard deviation (SD). AA1: mandibular arch; AA3: first branchial arch; AA4: second branchial arch; AA5: third branchial arch; AA6: fourth branchial arch; Ao: aorta; BA: bulbus arteriosus; cb1-4: ceratobranchial pairs 1 to 4; ch: ceratohyal; CO: cardiac output; dlav: dorsal longitudinal anastomotic vessel; et: ethmoid plate; h: hyosymplectic; HR: heart rate; isv: intersegmental vessel; m: Meckel's cartilage; ORA: opercular artery; pcv: posterior caudal vein; pq, palatoquadrate; SV: stroke volume; VA: ventral aorta. Scale bar = 200 μm (A), scale bar = 100 μm (E and G).

were of a uniform type characterized by mono- or disialylated Gal1-3Gal1-4(Fuc1-3)GlcNAc units according to MALDI-MS/MS. There was a significant increase in abundancy of oligo-mannose N-glycans, more specifically containing 9 mannose residues in *atp6v1e1b*-deficient larvae compared to wild-type (WT) controls (**Fig 2F**). We also observed the presence of differ-ent complex N-glycans in *atp6v1e1b*cmg78/cmg78 larvae which were absent in WT controls and in *atp6v1e1b*hi577aTg/hi577aTg larvae (**Fig 2G**). *Atp6v1e1b*cmg78/cmg78 contained preferentially

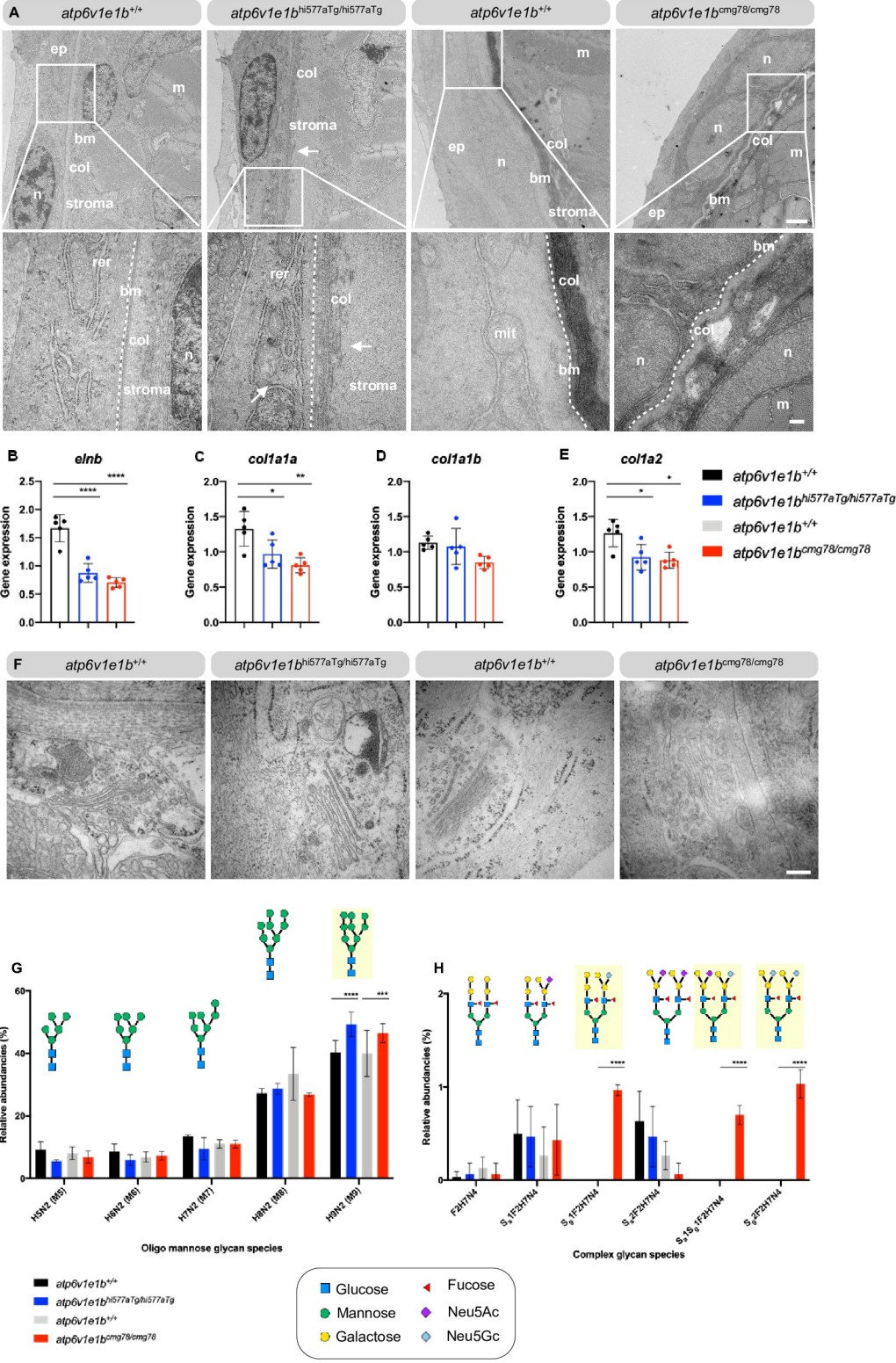

**Fig 2. Profound epidermal and N-glycosylation alterations in *atp6v1e1b*-deficient zebrafish.** (A) Representative images of ultrathin sections taken from the dermis of *atp6v1e1b*-deficient zebrafish and WT controls at 4 dpf. Note the two-layered epidermis that is separated from the collagenous stroma of the dermis by a well-defined basement membrane (bm), indicated as dotted line, at this developmental timepoint. In *atp6v1e1b*-deficient larvae, we observed a larger and more disorganized collagenous stroma of the dermis (*atp6v1e1b*^hi577aTg/hi577aTg^) or folded bm (*atp6v1e1b*^cmg78/

cmg78). Reproducible results were obtained in three independent experiments. Scale bar = 1 µm (top panel), scale bar = 200 nm (bottom panel). Bm: basement membrane; col: collagenous fibrils; ep: epidermis; m: muscle cell surface; mit: mitochondria; n: nucleus; stroma: primary dermal stroma; rer: rough endoplasmic reticulum. (B-E) Quantification of *elnb*, *col1a1a*, *col1a1b*, *col1a2* by RT-qPCR of *atp6v1e1b*-deficient larvae at 4 dpf compared to WT controls. Data are expressed as mean ± SD from 5 biological replicates. (F) Representative images demonstrating dilated Golgi apparatus in *atp6v1e1b*<sup>cmg78/cmg78</sup>. Reproducible results were obtained in three independent experiments. Scale bar = 200 nm (G) Quantification of relative abundancies of MALDI-TOF-MS signals for oligomannose N-glycan species detected in deyolked lysates in *atp6v1e1b*-deficient zebrafish compared to WT controls. (H) Quantification of relative abundancies of MALDI-TOF-MS signals for complex N-glycan species detected in deyolked lysates in *atp6v1e1b*-deficient zebrafish compared to WT controls. Data are expressed as mean ± SD from 3 biological replicates. 2-way ANOVA with Tukey test for multiple comparison was used for statistical analysis. Yellow shading indicates differences between WT controls and *atp6v1e1b*-deficient larvae. Symbols represent monosaccharide residues. Graphical representation is based on the accepted convention from the Symbol Nomenclature for Glycans (Consortium for Functional Glycomics). Yellow circle: Gal; green circle: Man; red triangle: Neu5Ac; light blue diamond: Neu5Gc; Neu5Ac: N-acetylneuraminic acid; Neu5Gc: N-glycolylneuraminic acid.

Neu5Gc, while *atp6v1e1b*<sup>hi577aTg/hi577aTg</sup> exclusively had Neu5Ac as sialic acid species. Upon TEM analysis, we observed a lower number of Golgi apparatus in *atp6v1e1b*-deficient zebrafish compared to WT controls that may be due to a high degree of Golgi apparatus fragmentation in *atp6v1e1b*-deficient zebrafish. The remaining Golgi apparatus in *atp6v1e1b*<sup>hi577aTg/hi577aTg</sup> are normal, but show a dilated phenotype in *atp6v1e1b*<sup>cmg78/cmg78</sup>.

## Loss of *atp6v1e1b* in zebrafish alters mTOR and Wnt signaling pathways *in vivo*

*Atp6v1e1b*-deficient zebrafish show increased protein levels of the early endosomal markers early endosome antigen 1 (EEA1) and small rab GTPase 5 (Rab5) compared with WT controls at 4 dpf. The late endosomal marker small rab GTPase 7 (Rab7) and the lysosomal glycoprotein marker, LAMP1, on the other hand tended to show lower protein levels in *atp6v1e1b*-deficient zebrafish (**S3A–S3F Fig**). Endosomal and lysosomal organelles have reduced acidification upon *atp6v1e1b* depletion in zebrafish, as shown by LysoTracker staining in the cranium (**S3G Fig**). Moreover, confocal images of endo(lyso)somal markers in tissues with highest expression levels show reduced Rab5 expression in the yolk (**S4A–S4D Fig**) and brain (**S4E–S4H Fig**) but accumulation of Rab7 in the muscle of the trunk region of *atp6v1e1b*<sup>hi577aTg/hi577aTg</sup>. Interestingly, we observed an inverse Rab5:Rab7 ratio with accumulation of Rab5 in the yolk and brain and decreased expression of Rab7 in the muscle of *atp6v1e1b*<sup>cmg78/cmg78</sup>. (**S4I–S4L Fig**). LAMP1 expression in the yolk (**S4M–S4P Fig**) and the brain (**S4Q–S4T Fig**) are normal between *atp6v1e1b*-deficient zebrafish compared to WT controls. Taken together, these observations indicate that loss of *atp6v1e1b* in zebrafish alters normal processing of endo(lyso)somal vesicles most likely due to abnormal acidification.

We investigated the impact of *atp6v1e1b* deficiency on the non-canonical signaling pathways regulated by v-ATPase. The ribosomal protein S6 kinases (S6Ks) are well-established effectors of mTORC1 signaling [32,46–48]. We assessed the (phosphorylated) protein levels of p70-S6K1, p85-S6K1, and p31-S6K1 in *atp6v1e1b*-deficient zebrafish and WT controls at 4 dpf. We observed remarkable up-regulation in total p85-S6K1 protein level in both *atp6v1e1b*-deficient zebrafish. The ratio phosphorylated to non-phosphorylated p85-S6K1 protein level was not significantly changed between *atp6v1e1b*-deficient zebrafish and WT controls (**Fig 3A–3C**). A significant increase in total p31-S6K protein level was observed in *atp6v1e1b*<sup>hi577aTg/hi577aTg</sup>. The ratio of phosphorylated p31-S6K protein level to p31-S6K protein level was significantly increased in both *atp6v1e1b*-deficient zebrafish (**Fig 3F and 3G**). A trend was noted toward decreased levels of total p70-S6K1 protein levels in both *atp6v1e1b*-deficient zebrafish, which did not reach statistical significance. Due to the large variability of phospho-p70-S6K protein levels within and between WT

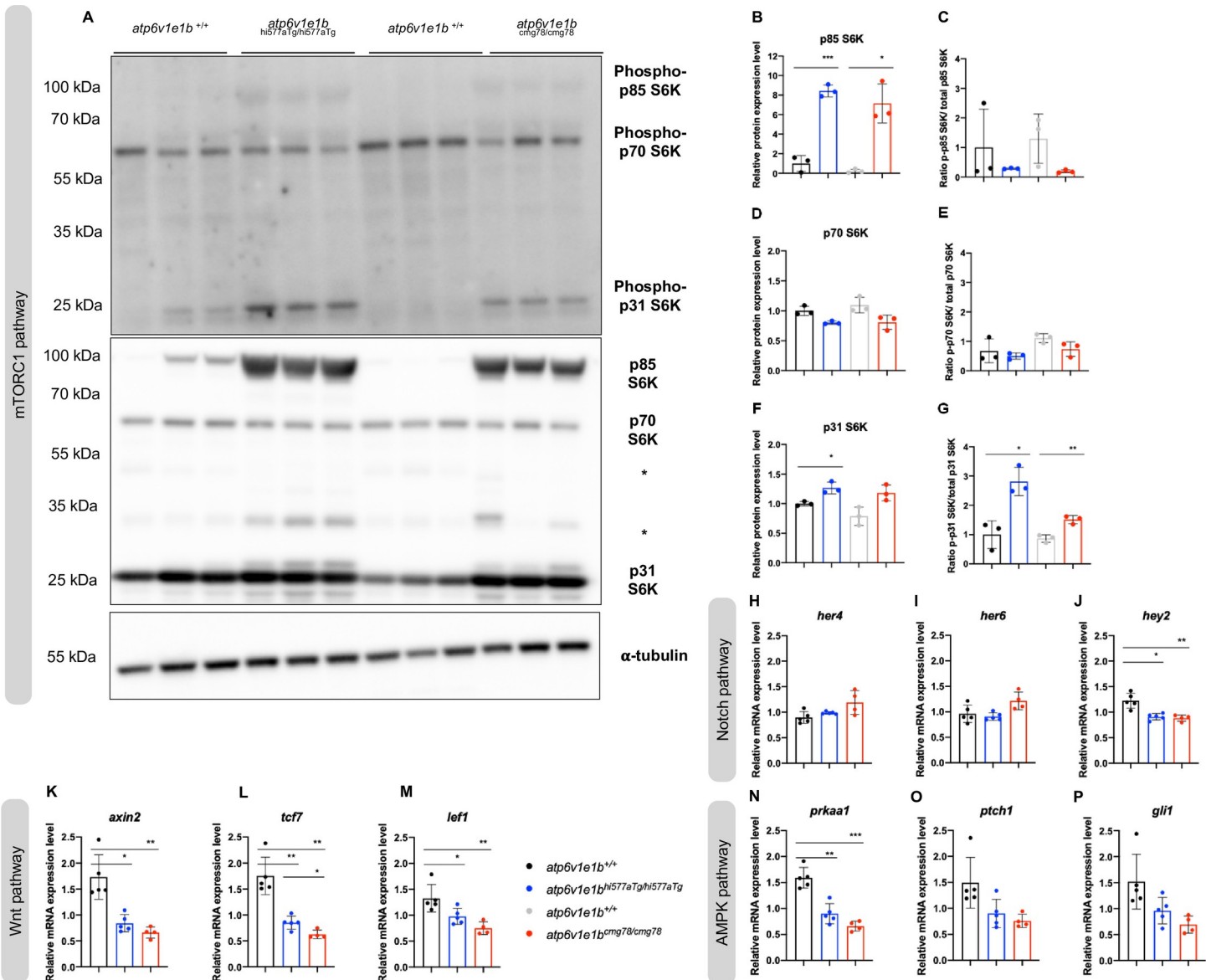

**Fig 3. Effect of loss of *atp6v1e1b* on the non-canonical function of v-ATPases.** (A) Phosphorylated ribosomal protein S6 kinase 1 (S6K1) (upper panel) and total S6K1 (lower panel) were detected in protein lysates of whole *atp6v1e1b*-deficient zebrafish and their corresponding WT controls at 4 dpf by immunoblotting. Asterisks indicate non-related, cross-reactive bands. Representative immunoblots are shown. We confirmed equal loading by staining for α-tubulin. (B-D-F) Band intensities of chemiluminescent signals of non-phosphorylated proteins were quantified with ImageJ and normalized to WT controls. (C-E-G) Band intensities of phosphorylated proteins were quantified with ImageJ. The ratio of phospho-p85-S6K to total p85-S6K, phospho-p70-S6K to total p70-S6K, and phospho-p31-S6K to total p31-S6K was normalized to WT controls. Data are expressed as mean ± SD from 3 biological replicates in B-G. (H-P) Gene expression of Notch-target genes (*her4*, *her6*, and *hey2*), Wnt-target genes (*axin2*, *tcf7*, and *lef1*), and AMPK-target genes (*prkaa1*, *ptch1*, and *gli1*) were investigated by RT-qPCR. Data are expressed as mean ± SD from 5 biological replicates in H-P.

control lines, the ratio of phospho-p70-S6K/ p70-S6K protein level remained stable between the different conditions (**Fig 3D and 3E**). As mTORC1 and AMPK signaling are interconnected, we next sought to determine the effect on AMPK-target genes. *Gli1*, *ptch1*, and *prkaa1* gene expression levels are downregulated indicating reduced AMPK activity (**Fig 3N**) [49]. In addition, Wnt signaling target genes (*axin2*, *tcf7*, and *lef1*) were significantly reduced in *atp6v1e1b*-deficient zebrafish at 3 dpf. In contrast, the majority (*her4*, *her6*) of the Notch signaling target genes (*her4*,

*her6*, *hey2*) remained unaltered compared to WT controls. Previous *in vitro* work has highlighted that decreased endo(lyso)somal acidification leads to reduced cytosolic bioavailability of iron and subsequent induction of HIF-1α signaling. HIF-1α-target genes (*egln3*, *vegfaa*, *vegfab*, *slc2a1a*, *slc2a1b*, *pfkfb3*, *angptl4* and *pdk1*) were downregulated in *atp6v1e1b*-deficient zebrafish at 3 dpf (**S5A–S5H Fig**). Moreover, E3 medium supplemented with Ferric ($Fe^{3+}$) ammonium citrate (FAC), which bypasses the endo(lyso)somal pathway through iron transporters in the plasma membrane [50], did not ameliorate the survival of *atp6v1e1b*-deficient zebrafish (**S5I Fig**). Also, hematopoiesis was not altered in *atp6v1e1b*-deficient zebrafish, as shown by o-dianisidine staining, providing further evidence of a normal availability of transferrin, hemoglobin and iron (**S5J Fig**). Taken together, we show that loss of *atp6v1e1b in vivo* does not stimulate the HIF-1α signaling pathways (in contrast to previous *in vitro* work) and mildly affects the Notch signaling pathway, but does have an impact on mTORC1, AMPK and Wnt signaling.

## Complex transcriptomic and metabolomic signature of loss of *atp6v1e1b in vivo*

We performed an unbiased assessment of the transcriptional and metabolic response to the loss of *atp6v1e1b* at 3 dpf, before gross morphological phenotypes become apparent, in order to attempt to identify the primary mechanisms underlying the physiological manifestations of v-ATPase dysfunction. Analysis of the transcriptomic data showed 112 significantly upregulated differentially expressed genes (DEGs) and 35 significantly downregulated DEGs in *atp6v1e1b*[hi577aTg/hi577aTg] (**Fig 4A**). In addition, in *atp6v1e1b*[cmg78/cmg78] 139 significantly upregulated DEGs and 29 significantly downregulated DEGs were found (**Fig 4B**). The top 30 most DEGs of *atp6v1e1b*-deficient zebrafish compared to WT controls are plotted in **Fig 4C**. We validated the expression levels of 10 genes picked from the top 150 most DEGs (**S6A Fig**). Generally applicable gene-set enrichment (GAGE) for pathway analysis using the KEGG pathway database identified several altered pathways in *atp6v1e1b*[hi577aTg/hi577aTg] and *atp6v1e1b*[cmg78/cmg78] highlighting oxidative phosphorylation, fatty acid elongation, carbon metabolism, and steroid biosynthesis (**Fig 4D**). Several genes in the top 30 most DEGs are linked to the observed phenotype via GAGE and reactome pathway analysis (**S6B and S6C Fig**).

To complement the gene expression studies, we assessed the metabolomic profiles of *atp6v1e1b*-deficient zebrafish in order to uncover non-genomic changes in enzymatic activity of a range of cellular and metabolic processes. From an untargeted metabolic screen, we evaluated pathway-level enrichments based on significant m/z peaks. Gene Set Enrichment Analysis (GSEA) metabolic pathway activity profiles are plotted in **Fig 4E** and **4F** highlighting positive associations for the lipid metabolism, fatty acid metabolism, glycolysis and gluconeogenesis, and negative associations for the glycosaminoglycans. Though identification of a given peak based on its mass alone is challenging, we found two compounds significantly downregulated that, based on the accurate mass, could correspond to xanthopterin and hydroxymethylpterin. These metabolites are related to pigmentation which is absent in *atp6v1e1b*-deficient zebrafish. Our transcriptome and metabolome data show that multiple pathways are affected, particularly with prominent alterations in several parts of energy metabolism and oxygen consumption.

## Significant upregulation of sphingolipids, phospholipids and dihydroceramide upon *atp6v1e1b*-deficiency in zebrafish

We found a striking accumulation of electron-dense vesicular bodies in the yolk of the mutant zebrafish (**Fig 5A**). These organelles were packed with membranous, lamellar lipid-like material.

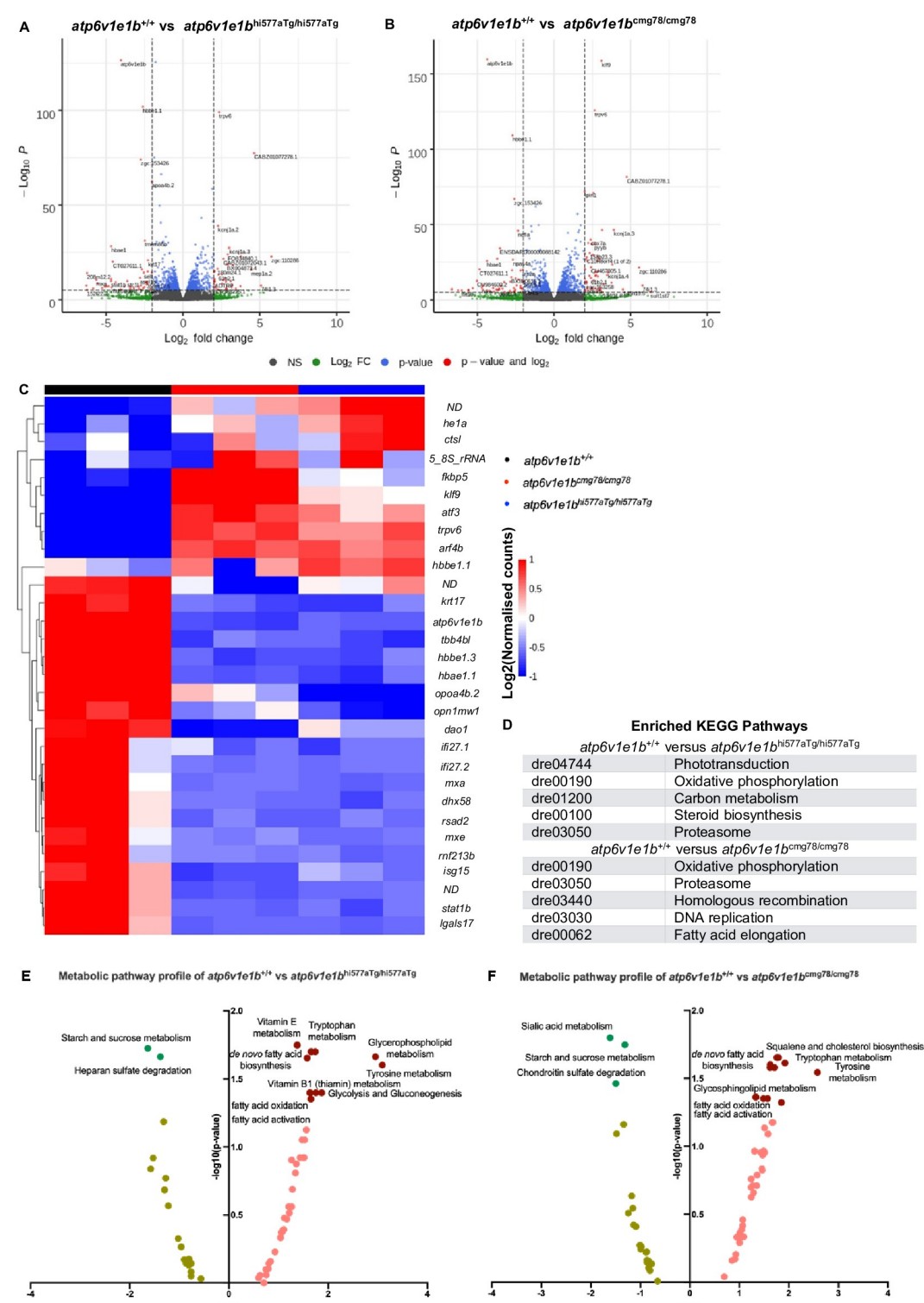

**Fig 4. Transcriptomic and metabolomic signature of *atp6v1e1b* deficiency *in vivo*.** (A) Volcano plot of RNA-sequencing data of *atp6v1e1b*^hi577aTg/hi577aTg. (B) Volcano plot of RNA-sequencing data of *atp6v1e1b*^cmg78/cmg78. The data represents overlapping genes from both ZFIN and Ensembl reference genome mapping (GRCz11). Red spots represent differentially expressed genes (DEGs). The horizontal line denotes the significance threshold (adjusted p<0.05) for DEGs. The vertical line denotes the Log$_2$ fold change (Fc) threshold of 2. (C) Hierarchical clustering of the top 30 most DEGs from whole-body samples of *atp6v1e1b*-deficient and WT control zebrafish at 3 dpf. DEGs are annotated with gene names of which the protein

function is known. DEGs with unknown gene names are listed in **S1 Table**. Colors range from red (high expression) to blue (low expression). (D) Results of the KEGG pathway analysis showing the most enriched pathways in the differentially expressed gene list of the whole-body samples from the *atp6v1e1b*-deficient and WT control zebrafish dataset. (E-F) Plots of an integrated analysis based on Metaboanalyst software (pathway tool) for a simplified view of contributing pathways in whole-body samples of *atp6v1e1b*-deficient zebrafish at 3 dpf. Full analysis based on GSEA parameters and Mummichog parameters are shown in **S2 Table**. Dark green and brown symbols represent significantly enriched pathways. ND: not determined; NES: normalized enrichment score.

*Atp6v1e1b*$^{\text{hi577aTg/hi577aTg}}$ contained larger membranous whorls than *atp6v1e1b*$^{\text{cmg78/cmg78}}$, in which multiple smaller membranous lipid whorls appeared in near proximity to each other (**Fig 5A**). Together with our -omics results, this encouraged us to study 19 different lipid classes using HILIC LC-MS/MS lipidome analysis to determine the consequences of *atp6v1e1b*-deficiency at 3 dpf. Phosphatidylcholine (PC), and its derivatives 1-alkyl,2- acylphosphatidylcholine (PC-O), 1-alkenyl,2-acylphosphatidylcholine (PC-P), lysophosphatidylcholine (LPC), as well as phosphati-dylglycerol (PG), phosphatidylinositol (PI), and sphingomyelin (SM) were significantly increased in *atp6v1e1b*-deficient zebrafish compared to WT controls (**Fig 5B–5H**). Ceramides (CER) were not significantly altered in *atp6v1e1b*-deficient zebrafish compared to the WT controls. In contrast, dihydroceramide (DCER), the precursor of CER via the *de novo* ceramide synthesis pathway located in the ER, is significantly increased. Hexosylceramide (HexCer) and lactosylceramide (LaxCer) were significantly decreased in *atp6v1e1b*-deficient zebrafish compared to WT controls (**Fig 5I–5L**). Levels of phosphatidylethanolamine (PE) and the related 1- alkyl,2-acylphosphatidy-lethanolamines (PE-O), 1-alkenyl, 2-acylphosphatidylethanolamines (PE-P) and lysophosphatidy-lethanolamine (LPE) showed a trend to be upregulated in *atp6v1e1b*-deficient zebrafish compared to WT controls (**S7A–S7D Fig**). Similarly, fatty acids such as triacylglycerides (TG), diacylglycer-ides (DG), cholesterol esters (CE), and cholesterol (Chol) showed some increases (**S7E and S7F Fig**). Inhibiting ceramide synthase, an enzyme that generates DCER in the *de novo* biosynthesis pathway in the ER, by administering Fumonisin B1 to the E3 medium at 1 dpf did not ameliorate the survival of *atp6v1e1b*-deficient zebrafish. The same observation was made when the ceramide transport protein (CERT), involved in the non-vesicular transport of ceramide from the ER membranes to Golgi apparatus for further processing to sphingomyelins, was inhibited by HPA-12 (**S8A–S8C Fig**). Also, nicotinic acid, an inhibitor for the cholesterol esterification was not able to induce beneficial effects. These data suggest that targeting only this pathway is not sufficient to rescue the pleiotropic consequences of *atp6v1e1b* deficiency *in vivo*.

## Defective oxygen consumption in *atp6v1e1b*-deficient larvae

Upon TEM examination, we observed dilated mitochondria in *atp6v1e1b*$^{\text{hi577aTg/hi577aTg}}$ and *atp6v1e1b*$^{\text{cmg78/cmg78}}$ zebrafish. In combination with our transcriptome data this observation suggests the existence of potential respiratory chain defects (**Fig 6A**), which prompted us to investigate the function of *atp6v1e1b*-deficient mitochondria. Oxygen consumption rate (OCR) measurements in WT controls, *atp6v1e1b*$^{\text{hi577aTg/hi577aTg}}$ and *atp6v1e1b*$^{\text{cmg78/cmg78}}$ zeb-rafish larvae at 2 dpf showed significantly decreased basal respiration in *atp6v1e1b*-deficient zebrafish compared to WT controls (**Fig 6B**). In addition, the basal extracellular acidification rate (ECAR) was lower in *atp6v1e1b*-deficient zebrafish compared to WT controls (**Fig 6C**). Measurement of OCR and ECAR estimates the ATP generation from two different pathways: oxidative phosphorylation in the mitochondria and glycolysis in the cytosol [51]. Plotting ECAR against OCR of both mutant zebrafish and WT controls showed an altered metabolic state. *Atp6v1e1b*-deficient zebrafish showed a quiescent signature, while the corresponding WT controls are in a more energetic state (**Fig 6D**). Administration of the proton uncoupler carbonyl cyanide p-trifluoromethoxy- phenylhydrazone (FCCP) (**Fig 6F–6H**) had no

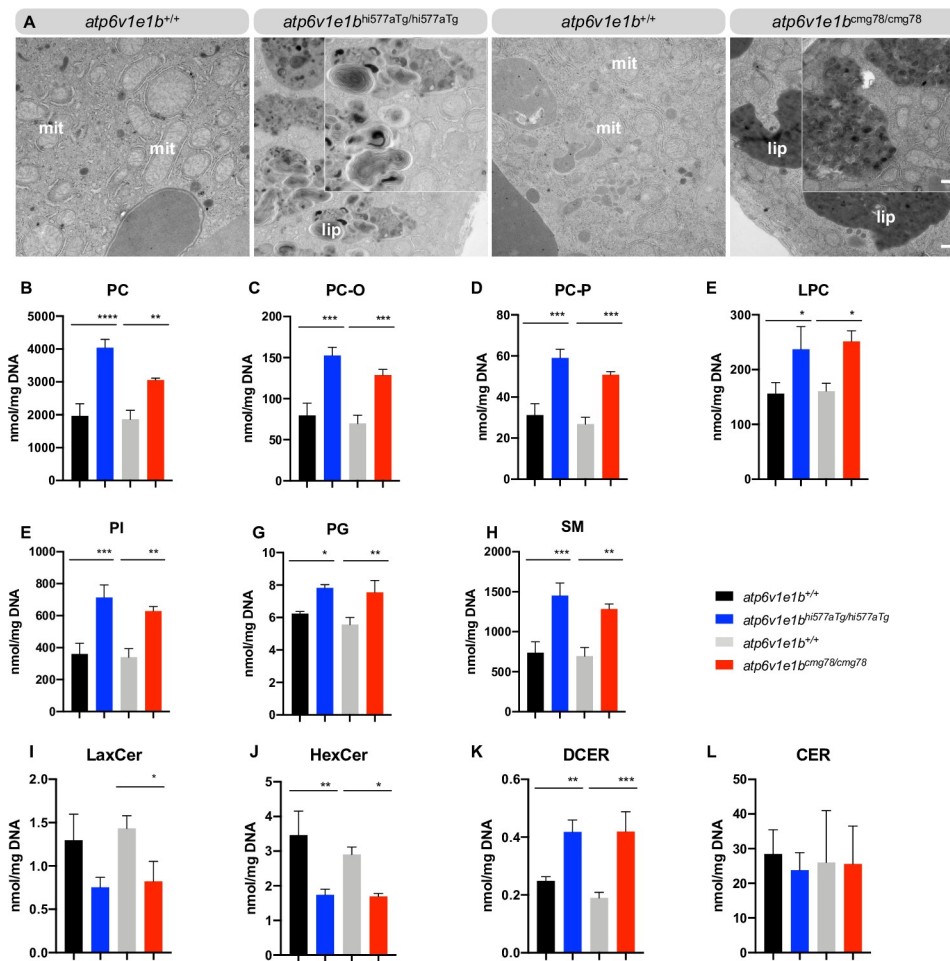

**Fig 5. A*tp6v1e1b* depletion leads to higher amounts of sphingolipids and phospholipids in zebrafish larvae.** (A) Representative images of ultrathin sections of the yolk from 4 dpf WT control and *atp6v1e1b*-deficient zebrafish. *Atp6v1e1b*-deficient larvae reveal an accumulation of electron-dense vesicular bodies. Higher magnification of the lipid whorls is shown in the top right corner of the corresponding image. Scale bar = 500 nm (low magnification), scale bar = 200 nm (high magnification). Results are representative of three independent experiments. Mit: mitochondria; endo: endosomal derived multilamellar bodies; lip: lipid whorls. (B-L) HILIC LC-MS/MS lipidomic analysis of *atp6v1e1b*-deficient zebrafish and WT controls at 3 dpf. Data are expressed as mean ± SD from 3 biological replicates. PC: phosphatidylcholine; PC-O: 1-alkyl,2-acylphosphatidylcholine; PC-P: 1- alkenyl,2-acylphosphatidylcholine; LPC: lysophosphatidylcholine; PG: phosphatidylglycerol; PI: phosphatidylinositol; SM: sphingomyelin; CER: ceramides; DCER: dihydroceramides; HexCer: hexosylceramides; LacCer: lactosylceramides.

influence on the OCR in WT controls. FCCP however did increase the OCR in *atp6v1e1b*-deficient zebrafish, indicating spare respiratory capacity (**Fig 6F–6H**). Administering oligomycin, a compound that inhibits the mitochondrial F-ATP synthase (complex V of the oxidative respiratory chain) [52], decreases the mitochondrial respiration which is linked to cellular ATP production in WT controls. Surprisingly, we observed an immediate increase in OCR directly after administration of oligomycin in both *atp6v1e1b*-deficient zebrafish (**Fig 6E–6G**). Administration of Bafilomycin A1 (Baf A1) to WT controls reduced the mitochondrial oxygen consumption which suggests a direct link between altered v-ATPases and mitochondrial respiration in zebrafish (**S9B–S9D Fig**). Baf A1 treatment also reduced the mitochondrial oxygen consumption in *atp6v1e1b*^hi577aTg/hi577aTg^ (**S9B Fig**). In *atp6v1e1b*^cmg78/cmg78^, Baf A1 treatment had no effect on basal respiration (**S9D Fig**). Interestingly, the immediate increase in OCR

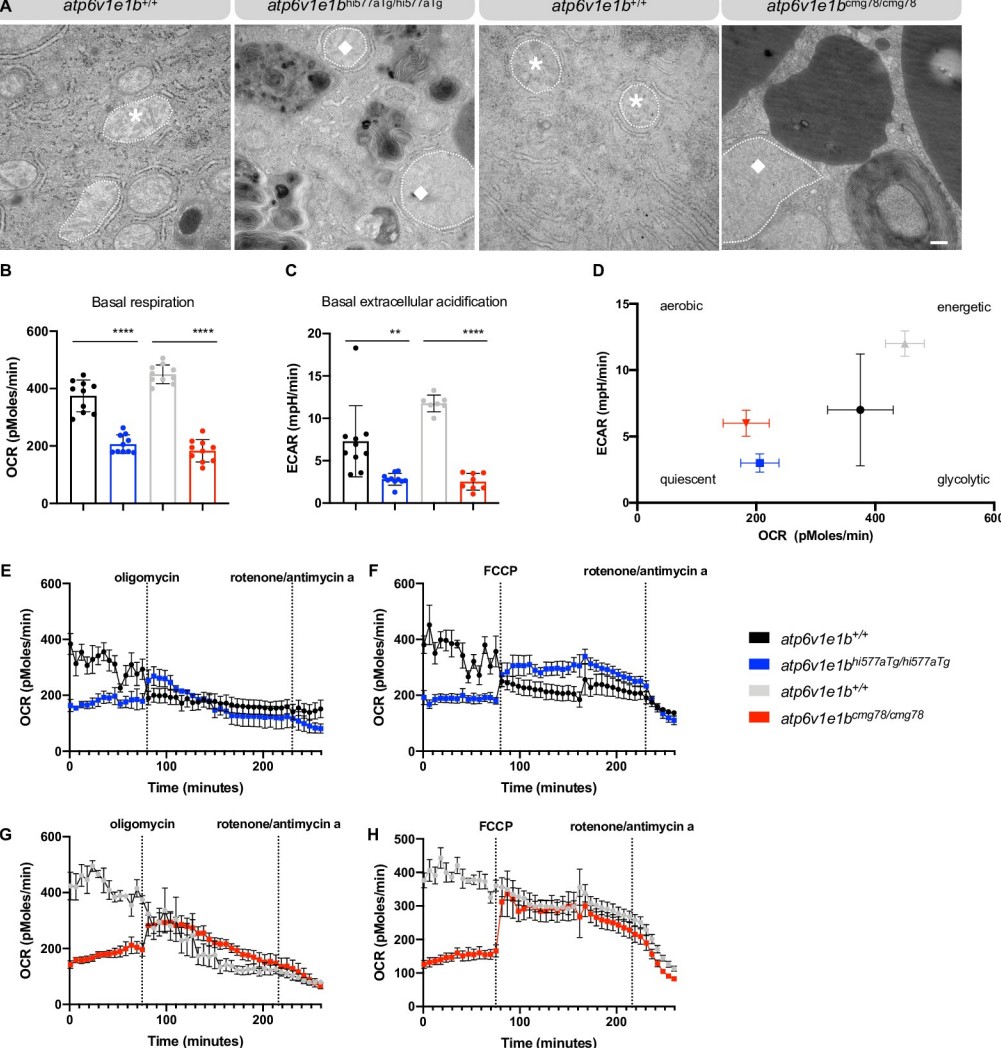

**Fig 6. Early respiratory chain deficits resulting from *atp6v1e1b* depletion *in vivo*.** (A) Representative images of ultrathin sections of the yolk from WT control and *atp6v1e1b*-deficient zebrafish and WT controls at 4 dpf. Asterisk, normal mitochondria. Diamond, dilated mitochondria. Scale bar = 500 nm. Results are representative of three independent experiments. (B) Measurement of the basal respiration and (C) extracellular acidification rate in *atp6v1e1b*<sup>hi577aTg/hi577aTg</sup> and *atp6v1e1b*<sup>cmg78/cmg78</sup> zebrafish larvae at 2 dpf compared to WT controls. (D) Plot of OCR vs. ECAR values showed that both mutant zebrafish larvae are in a more quiescent aerobic state than their WTs. (E-H) OCR in WT control and *atp6v1e1b*-deficient zebrafish after administration of oligomycin and rotenone/antimycin a or FCCP and rotenone/antimycin a. ECAR: extracellular acidification rate; OCR: oxygen consumption rate. Data are expressed as mean ± standard error of the mean (SEM) from 5 biological replicates.

directly after administration of oligomycin in both *atp6v1e1b*-deficient zebrafish remained present after administration of Baf A1 (**S9A–S9D Fig**). Considering the observed metabolic abnormalities in the *atp6v1e1b*-deficient zebrafish, we supplemented metabolites of the Krebs cycle to the E3 medium of mutant larvae in order to attempt to ameliorate their phenotype. Administration of pyruvate, fumarate, oxaloacetic acid, succinate, glycolate, and L-(-)-acetic acid was not able to improve the survival of *atp6v1e1b*-deficient zebrafish (**S8D–S8I Fig**). This indicates that targeting the Krebs cycle alone is not sufficient to rescue the pleiotropic consequences of reduced *atp6v1e1b in vivo*.

## Discussion

In order to investigate the molecular mechanisms involved in the pathogenesis of ARCL type 2C, which is caused by pathogenic variants in the *ATP6V1E1* gene [10], we performed an in-depth characterization of genetically modified zebrafish models. Since zebrafish have two ohnologs of *ATP6V1E1*, we investigated both *atp6v1e1a-* and *atp6v1e1b*-deficient zebrafish. While zebrafish deficient in *atp6v1e1a* survived normally to adulthood without any phenotype, a*tp6v1e1b*-deficiency led to embryonical lethality, associated with muscular, craniofacial, cardiovascular, and dermal abnormalities. These data confirm the pleiotropic effects of v-ATPase deficiency, mimicking the syndromic manifestations of ARCL type 2C patients. Moreover, our data suggest that atp6v1e1a is a non-functional ohnolog, however unrecognized compensatory mechanisms might still be discovered in future evolutionary studies.

Based on the known N-glycosylation abnormalities in ARCL type 2C patients, we characterized the glycosylation profile in the *atp6v1e1b*-deficient zebrafish. 9-mannose N-glycan species were increased in both *atp6v1e1b*-deficient strains, but differences in complex N-glycan species were more pronounced in *atp6v1e1b*$^{cmg78/cmg78}$ than in *atp6v1e1b*$^{hi577aTg/hi577aTg}$ zebrafish. In *atp6v1e1b*$^{cmg78/cmg78}$ larvae, a higher abundance of Neu5Gc terminal glycan species (instead of Neu5Ac terminal glycan species) likely reflects altered activity of cytidine monophospho-N-acetylneuraminic acid hydroxylase (CMAH) [53]. Defective N-glycosylation is likely associated with the reduced abundance of Golgi apparati in both *atp6v1e1b*-deficient zebrafish strains. Even more, the remaining Golgi showed a dilated phenotype in *atp6v1e1b*$^{cmg78/cmg78}$, but not in *atp6v1e1b*$^{hi577aTg/hi577aTg}$ larvae, which could point to the stronger disruption of sequential enzymatic N-linked glycosylation modifications [54]. Reduced acidification in the *cis*-Golgi might cause suboptimal enzymatic functioning of mannosidase 1 altering trimming of 9-mannose N-glycans towards subsequent substrates for further processing in the medial-Golgi [54–56]. While the difference in the complex glycan profiles between both mutant lines is interesting from a mechanistic perspective, the relevance for human biology is likely limited since the CMAH gene has been inactivated during evolution just prior to human speciation [53]. The exact mechanisms underlying the differences between the two mutant zebrafish lines remain to be elucidated in future studies, but they likely involve residual atp6v1e1b protein activity still being expressed in the *atp6v1e1b*$^{hi577aTg/hi577aTg}$ line [57]. Taken together, these features are reminiscent to the changes in relative abundancies of different N-glycans and altered Golgi morphology observed in ARCL type 2C patients [10].

Early endosomes are known to mature into late endosomes by Rab conversion [58]. Different tissue-specific expression patterns of Rab5 and Rab7 in *atp6v1e1b*-deficient zebrafish larvae suggest altered processing of endosomes correlating with residual atp6v1e1 protein level. In addition, our Lysotracker data suggests that loss of *atp6v1e1b* in zebrafish led to an increase in the pH in acidic vesicles due to the loss of v-ATPase functionality. Further studies should confirm the increase in the pH of acidic vesicles by creating transgenic zebrafish lines expressing the pHluorin (pH-sensitive GFP variant pHluorin) and pHLARE (pH Activity Reporter based on fusion with LAMP1) reporters [59,60] since measuring pH in zebrafish is not straightforward. *In vitro*, it has been shown that inhibition of v-ATPases by siRNAs or by specific inhibitors leads to a decrease in the acidification of endosomes and lysosomes, blocking the activity of lysosomal proteases. As a result, endosome maturation and processing as well as lysosomal protein degradation is disrupted. Subsequent decreases in intra-lysosomal amino acid levels were shown to reduce mTORC1 activity [61–63]. In contrast, we found that loss of *atp6v1e1b* in zebrafish larvae induces an increase in total protein levels of the cytoplasmic p85 isoform of S6K1 along with an increase in the nuclear p31 isoform of S6K1. The latter may indicate that splicing protein SF2/ASF, known to induce oncogenic activity *in vitro*, might be

active in *atp6v1e1b*-deficient zebrafish larvae [47,64]. Nevertheless, an increased mTORC1 activity has previously been observed in primary osteoclasts derived from mice carrying a point mutation in the *Atp6v0a3* causing increased lysosomal pH [65–67]. In parallel, our data also shows decreased AMPK activity in *atp6v1e1b*-deficient zebrafish. Taken together, the modulation of both critical sensors suggests that loss of *atp6v1e1b* and reduced vesicular acidification directs cells towards an anabolic response.

We furthermore investigated the non-canonical effects of v-ATPase deficiency on a number of signalling pathways. We found reduced gene expression levels of canonical Wnt target genes in *atp6v1e1b*-deficient zebrafish, likely due to a reduced signalling from the Wnt-receptor complex (LRP6, PRR, and Wnt receptor) which requires activation by v-ATPase-dependent acidification in endosomes [34,68]. On the other hand, no prominent disruption of the Notch signalling pathway could be shown in our *atp6v1e1b*-deficient zebrafish models, despite evidence from prior studies indicating that endosomal acidification is required for Notch signal transduction [69].

Intriguingly, we discovered lipid-rich multilamellar bodies, characterized as lipid whorls, in the yolk of *atp6v1e1b*-deficient zebrafish. Lipid whorls are also observed in primary and secondary lysosomal storage disorders such as Niemann–Pick disease type C (NPC), which are characterized by reduced substrate clearance, and an accumulation of endo(lyso)somes [70,71]. In NPC, deficiency of the NPC1 or NPC2 protein impairs the transmembrane transport of lipids, especially cholesterol, in the late endo(lyso)somal pathway [72]. It has previously been shown that v-ATPase subunits are associated with detergent resistant membranes present in late endosomes containing raft-like domains which are rich in cholesterol and sphingomyelin [73–75]. Our data confirmed a strong increase of PC levels and its derivatives, PG, PI, and SM levels in *atp6v1e1b*-deficient zebrafish. However, DCER levels are increased while CER levels remain equal. Since *atp6v1e1b*-deficient zebrafish show reduced and abnormal Golgi, we hypothesize that translocation of ceramides might be reduced or delayed, leading to accumulation of CER and its direct precursor DCER. The overall amount of CER in the embryo is unaltered, likely as a result of coupling between the *de novo* and salvage pathways and action of sphingomyelinase at the plasma membrane [76]. Further studies are required to fully elucidate the mechanisms responsible for the lipid profile abnormalities observed in the two mutant zebrafish lines.

To better understand the system-wide effects of v-ATPase dysfunction *in vivo*, we performed an unbiased analysis of the transcriptome and metabolome in *atp6v1e1b*-depleted zebrafish. These assays were performed at a developmental stage before gross morphological alterations became apparent, in an attempt to minimize signals from secondary effects resulting from general physiological dysregulation. KEGG pathway analysis on DEGs in *atp6v1e1b*-deficient zebrafish indicated enrichment of genes involved in oxidative phosphorylation, fatty acid elongation, carbon metabolism, and steroid biosynthesis pathways. In addition, unbiased analysis of the metabolic profile of *atp6v1e1b*-deficient zebrafish indicated glycosphingolipid metabolism, glycerophospholipid metabolism, glycolysis and gluconeogenesis, and fatty acid metabolism as key affected pathways. Taken together, our systems biology approach has shown that perturbation of v-ATPase function *in vivo* results in complex changes in multiple biological pathways, with a marked impact on energy and lipid metabolism.

The system-wide indications of altered energy metabolism, together with the observation of dilated mitochondria, prompted us to further investigate the mitochondrial respiratory chain in our *atp6v1e1b* mutant models. Interestingly, we found that the uncoupling agent FCCP did not increase OCR in 2 dpf WT larvae beyond baseline levels, suggesting that mitochondria are already at a maximal respiration rate at this stage of development. In contrast, 2 dpf *atp6v1e1b*-deficient zebrafish larvae showed a decreased OCR at baseline, that could be increased

to similar levels as observed in WT larvae by the administration of FCCP, suggesting suboptimal functioning of the respiratory chain. Surprisingly, after administering the ATP synthase inhibitor oligomycin, we observed an unexpected increase in the OCR of *atp6v1e1b*-deficient zebrafish, while WT larvae showed an immediate decrease in OCR as expected. We hypothesize that this could be the result of a reverse direction of operation of the mitochondrial ATP synthase (F-ATPase) at baseline in *atp6v1e1b*-deficient zebrafish. ATP hydrolysis activity of F-ATPase has previously been observed *in vitro* when mitochondrial dysfunction is induced by disrupting the electrochemical membrane potential [77–79]. Given the evolutionary relationship between v-ATPase and F-ATPase, alteration of the E1 subunit of v-ATPase might directly impact F-ATPase function, a hypothesis supported by observations in trypanosomes, where v-ATPase depletion was shown to affect F-ATPase coupling [80]. Mitochondrial dysfunction is linked with reduced muscle functions [81]. We hypothesize that less mitochondrial ATP production might cause hypotonia in *atp6v1e1b*-deficient zebrafish. Administration of Baf A1 to 2 dpf WT embryos mimics the effects of *atp6v1e1b* deficiency on mitochondrial respiration and provides a direct link between v-ATPase inhibition and impaired mitochondrial respiration. These *in vivo* data confirm previous *in vitro* data obtained in mouse embryonal fibroblasts treated with Baf A1, showing an even more pronounced shutdown of mitochondrial respiration [36–38]. The decline in basal OCR after Baf A1 administration in *atp6v1e1b*[hi577aTg/hi577aTg] but not in *atp6v1e1b*[cmg78/cmg78] larvae is likely due to the presence of residual atp6v1e1 protein level in the former line. This newly discovered *in vivo* link between v-ATPase deficiency and mitochondrial dysfunction, likely plays an important role in the pathogenesis of disorders caused by affective v-ATPase function and merits more research into the specific mechanisms involved.

In line with the mechanistic evidence gathered in our studies, we aimed to test whether modulating the identified pathways could ameliorate the phenotype in *atp6v1e1b*-deficient zebrafish. Inhibition of ceramide synthase and ceramide transport protein did not have any effect on the survival of mutant larvae. Moreover, the administration of drugs interfering with the Krebs cycle did also not improve the survival of *atp6v1e1b*-deficient zebrafish. These results suggest that we might have not yet found the right therapeutic target, or that compounds modulating multiple targets or pathways will be necessary to rescue the consequences caused by *atp6v1e1b* deficiency. Future larger compound screens in these zebrafish models are likely to yield novel therapeutic targets which could potentially be pursued for further clinical applications.

A limitation of this study is that only whole-body zebrafish embryo samples were investigated, due to the technical difficulties to extract specific cell or tissue types at the early developmental stages. Therefore, it is conceivable that effects limited to specific cell types are obscured in our multi-omics data analysis. On the other hand, since homozygous *atp6v1e1*-deficient zebrafish are not viable, maternal zygotic knockout embryos cannot be obtained and the possibility remains that the phenotype during early development is mitigated by residual protein level from maternal WT *atp6v1e1b* mRNA in homozygous mutant embryos. In a prior study investigating the relative contribution of the maternal and paternal transcriptome in developing zebrafish embryos, RNAseq analysis indicated that *atp6v1e1b* mRNA was almost exclusively maternally derived up to 6 hpf, but starting at 9 hpf strong zygotic expression was detected [82]. These data suggest that maternally-derived *atp6v1e1b* transcripts only play a role during early embryogenesis, and that zygotic *atp6v1e1b* transcription takes over after gastrulation. Considering the expected minimal residual impact of maternally-derived WT *atp6v1e1b* transcript at the time of analysis and due to the profound presence of lipid alterations in the yolk we decided to study intact embryos in our omics experiments, thereby excluding potential artefactual disturbances to the measurements which might be introduced by a

deyolking procedure [83]. The only exception was the analysis of the N-glycosylation profile in embryos which does require prior deyolking in order to maximize the relevance of the obtained results. The signal from early embryonic body proteins that are present in smaller proportions might be obscured by the signal from the predominant yolk proteins [45,84]. In future studies cell- and/or tissue-specific inactivation of *atp6v1e1b* can be investigated in order to further explore the detailed mechanisms underlying the wide range of manifestations of v-ATPase deficiency.

In summary, our mutant zebrafish models resemble the human ARCL type 2C phenotype with pleiotropic manifestations in multiple tissues. Unbiased analysis of the effects of loss of v-ATPase function on the level of the transcriptome, metabolome, glycome, and lipidome indicated alterations in oxidative phosphorylation, fatty acid metabolism, glycolysis profile, and sphingolipid metabolism as most prominently affected processes. Taken together, our study has identified a range of biological effects of *atp6v1e1b* deficiency *in vivo*, which deserve further investigation and might be leveraged to develop new therapeutics in the future.

## Materials and methods

### Ethics statement

We adhered to the general guidelines, in agreement with EU Directive 2010/63/EU for laboratory animals, for zebrafish handling, mating, embryo collection and maintenance [85,86]. Approval for this study was provided by the local committee on the Ethics of Animal Experiments (Ghent University Hospital, Ghent, Belgium; Permit Number: ECD 17/63K and ECD 18/05).

### Zebrafish lines and maintenance

Zebrafish lines were housed in a Zebtec semi-closed recirculation housing system at a constant temperature (27–28˚C), pH (~7.5), conductivity (~550 μS) and light/dark cycle (14/10). Fish were fed twice a day with dry food (Gemma Micro) (Skretting, France) and once with artemia (Ocean Nutrition, Essen, Belgium). *Atp6v1e1b*[hi577aTg/+] was purchased from the Zebrafish International Research Center (ZIRC, University of Oregon, Eugene, USA). *Atp6v1e1b*[hi577aTg/hi577aTg] constitutes a retroviral insertion in the 5' UTR of the gene [41,87]. *Atp6v1e1b*[cmg78/cmg78] and *atp6v1e1a*[-/-] were generated using CRISPR-Cas9 mutagenesis according to the workflow previously described [88]. Zebrafish were genotyped with primers listed in **S3 Table**.

### Alcian Blue staining

Cartilage patterns were stained with Alcian Blue as previously described [89]. Craniofacial structures were measured with Fiji software [90] and normalized to the length of the head. Stained specimens were analyzed with a Leica M165 FC Fluorescent Stereo Microscope (Leica Microsystems, GmbH, Wetzlar, Germany).

### Evaluation heart function and blood flow

Video microscopy was performed on an Axio Observer.Z1 inverted microscope attached to a monochrome Axiocam 506 camera (Carl Zeiss Microscopy GmbH, Jena, Germany). Sequential images of the heart and blood flow region of interest were obtained in lateral position at 50 frames/sec at 3 dpf. Videos of the heart were manually analyzed using Fiji software [90]. The short and long axis of an ellipse fitted to the outline of the outside ventricular wall was measured 3 times in systole and in diastole [91]. Based on the assumption that the ventricle of the zebrafish embryo is a prolate spheroid [92], the systolic and diastolic volume was calculated.

Videos of the blood flow through the tail vasculature were analyzed as previously described [93].

## Evaluation of the vasculature structure

Zebrafish were outcrossed to the *Tg(kdrl:eGFP)* zebrafish line in order to visualize endothelial cells. Zebrafish larvae of 5 dpf were embedded in 0.8% seaPlaque low melting agarose (Lonza, Basel, Switzerland) supplemented with 160 mg/L tricaine (Sigma-Aldrich, Saint Louis, MI, USA) in order to minimize the artefactual movements. Z-stacks were taken on an Axio Observer.Z1 inverted microscope attached to a monochrome Axiocam 506 camera (Carl Zeiss Microscopy GmbH) and analyzed using the stack focuser algorithm in Fiji software [90]. The segmental area of the aorta between the first (AA3) and the second branchial arch (AA4) [94], diameter of aorta proximal to bulbus, average of 3 segments between branches of the ventral aorta and the diameter of the bulbus arteriosus were measured.

## Transmission electron microscopy

Zebrafish mutant larvae of 4 dpf were fixed and processed for ultrastructural analysis as previously described [95]. Sections were viewed with Jeol JEM 1010 TEM (Jeol Ltd., Tokyo, Japan) equipped with a CCD side mounted Veleta camera operating at 60 kV. Experiments were performed in collaboration with the TEM facility of the Nematology Research Unit. Pictures were digitized using a Ditabis system (Ditabis Ltd., Pforzheim, Germany).

## Western blot

For immunoblot analysis, 20 zebrafish larvae were homogenized with a pestle in SDS Laemlli buffer (10% glycerol, 2% SDS, 0.5M Tris HCl pH 8,0 and dH$_2$O) at 4 dpf. Protein samples were subjected to 7%, 10% or 15% SDS polyacrylamide gel electrophoresis, and transferred to a polyvinyldifluoride (PVDF) or nitrocellulose (NC) membrane (Thermo Fisher Scientific, Waltham, Massachusetts, USA) with the iBlot 2 Dry Blotting System (Thermo Fisher Scientific). Imaging was performed with an Amersham Imager 600 CCD camera (GE Healthcare, Chicago, Illinois, USA) and analyzed using Fiji software [90]. The following primary antibodies were used: monoclonal rabbit anti-Rab5 antibody (1:1000, C8B1, Cell Signaling Technology (CST), Danvers, Massachusetts, USA), monoclonal rabbit anti-Rab7 antibody (1:1000, D95F2, CST), monoclonal rabbit anti-EEA1 antibody (1:1000, C45B10, CST), polyclonal rabbit anti-LAMP1 (1:1000, ab24170, Abcam, Cambridge, United Kingdom) and polyclonal rabbit anti-ATP6V1E1 (1:500, ab111733, Abcam). HRP-conjugated goat anti-rabbit IgG was used as a secondary antibody (1:2000, #7074, CST).

## RT-qPCR

Quantitative reverse transcription PCR (RT-qPCR) was performed as described previously [96]. Total RNA was extracted in quintuplicate, in which 10 zebrafish larvae were pooled per sample. Assays were prepared with the addition of ssoAdvanced SYBR Green supermix (Bio-Rad Laboratories, Hercules, CA, USA) and were subsequently run on a LightCycler 480 Instrument II (Roche, Basel, Switzerland). Primers were designed using Primer-BLAST (**S5 Table**). Biogazelle qBase+3.0 Software was used for data analysis using *bactin2*, *elfa* and *gapdh* for normalization.

## Oxygen measurements

An XF24 Extracellular Flux Analyzer (Agilent Technologies, Santa Clara, CA, USA) was used to measure oxygen consumption rate (OCR) and extracellular acidification rate (ECAR). 10 of

the 24 wells on an islet capture microplate (Agilent Technologies) were used for mutant zebra-fish larvae, 10 wells were used for their respective WT controls and 4 wells served as quality control (QC). Four zebrafish larvae were put together in 1 well. Zebrafish larvae were incubated in 1x E3-medium buffered with 4mM 4-(2-hydroxyethyl)-1-piperazineethanesulfonic acid (HEPES) (Life Technologies, Carlsbad, CA, USA). Seahorse XF Cell Mito Stress kit (Agilent Technologies) was used according to manufacturer's instructions and modified for zebrafish purposes as described in previous publications [97–99]. Baf A1 (1,6 μM final concentration, Sigma-Aldrich) was administered to the zebrafish larvae in order to investigate the effect of Baf A1 on the ORC response. Final DMSO concentration did not exceed 1% in the well of the islet capture microplate. OCR data and ECAR data were exported using Wave Desktop Software (Agilent Technologies).

## Lipid extraction, mass spectrometry and data analysis

40 larvae of 3 dpf were homogenized in 500 μL $H_2O$ with the Precellys system and a sample volume equal to 10 μg of DNA was diluted in 700 μL $H_2O$ and mixed with 800 μL 1 N HCl: CH3OH 1:8 (v/v), 900 μL $CHCl_3$ and 200 μg/ml of the antioxidant 2,6-di-tert-butyl-4-methyl-phenol (BHT) (Sigma Aldrich). 3 μL of SPLASH LIPIDOMIX Mass Spec Standard, (Avanti Polar Lipids, Alabaster, AL, USA) was spiked into the extraction mix. The organic fraction was evaporated using a Savant Speedvac spd111v (Thermo Fisher Scientific) at room temperature and the remaining lipid pellet was stored at -20˚C under argon. Just before mass spectrometry analysis, lipid pellets were reconstituted in 100% ethanol. Lipid species were analyzed by liquid chromatography electrospray ionization tandem mass spectrometry (LC-ESI/MS/MS) on a Nex-era X2 UHPLC system (Shimadzu, Kioto, Japan) coupled with hybrid triple quadrupole/linear ion trap mass spectrometer (6500+ QTRAP system; AB SCIEX). Chromatographic separation was performed on a XBridge amide column (150 mm × 4.6 mm, 3.5 μm; Waters) maintained at 35˚C using mobile phase A (1 mM ammonium acetate in water-acetonitrile 5:95 (v/v)) and mobile phase B (1 mM ammonium acetate in water-acetonitrile 50:50 (v/v)) in the following gradient: (0–6 min: 0% B ➔ 6% B; 6–10 min: 6% B ➔ 25% B; 10–11 min: 25% B ➔ 98% B; 11–13 min: 98% B ➔ 100% B; 13–19 min: 100% B; 19–24 min: 0% B) at a flow rate of 0.7 mL/min which was increased to 1.5 mL/min from 13 minutes onwards. SM, CE, CER, DCER, HCER, LCER were measured in positive ion mode with a precursor scan of 184.1, 369.4, 264.4, 266.4, 264.4 and 264.4 respectively. TAG, DAG and MAG were measured in positive ion mode with a neutral loss scan for one of the fatty acyl moieties. PC, LPC, PE, LPE, PG, LPG, PI, and LPI were measured in negative ion mode with a neutral loss scan for the fatty acyl moieties. Lipid quantification was performed by scheduled multiple reactions monitoring, the transitions being based on the neutral losses or the typical product ions as described above. Peak integration was performed with the MultiQuant software version 3.0.3. Lipid species signals were corrected for isotopic contributions (calculated with Python Molmass 2019.1.1) and were normalized to internal standard signals. Unpaired T-test p-values and FDR corrected p-values (using the Benjamini/Hochberg procedure) were calculated in Python StatsModels version 0.10.1.

## Transcriptomics

Total RNA was isolated from whole zebrafish larvae at 3 dpf from *atp6v1e1b*[hi577aTg/hi577aTg] and *atp6v1e1b*[cmg78/cmg78] zebrafish, and WT controls as described previously [96]. Total RNA was extracted in triplicate. RNA integrity was checked by 2100 Bioanalyzer (Agilent Technologies). 10 zebrafish larvae of each genotype were pooled in 1 sample to obtain sufficient hetero-geneity amongst samples. A sequencing library was prepared using the TruSeq Stranded mRNA Library Prep (Illumina, San Diego, CA, USA) supplemented with TruSeq RNA Single

Indexes Set A (Illumina), according to the manufacturer's instructions. Paired-end sequencing was performed on an HiSeq 3000 sequencer (Illumina) with the HiSeq 3000/4000 SBS kit (150 cycles) according to the manufacturer's protocol. Data analysis was done in collaboration with the bio-informatics core (Department of Biomolecular Medicine, UGent). An RNA-seq pipeline was used that was published by the nf-core community. This pipeline was executed using the Nextflow [100] engine for computational workflows and comprises several processing steps [101]. QC analysis of the RNA-seq data was performed with FastQC and MultiQC [102]. TrimGalore was used to remove adapter contamination and to trim low-quality regions. Duplicate reads were identified with MarkDuplicates (Picard). Subsequently, all cleaned and trimmed reads that passed QC were aligned to GRCz10 using STAR aligner [103]. The index trimmed paired-end 150 base pair reads were aligned to the zebrafish GRCz10 reference genome. Gene counts were computed using the featureCounts package [104]. Differential expression analysis subsequently was performed on these gene counts using DESeq2 [105]. Differentially expressed genes were identified using a fold change cut-off >1 and FDR = 0.05. Finally, GO enrichment, GO pathway (http://geneontology.org/) and KEGG pathway (https://www.genome.jp/kegg) analysis were performed on differentially expressed gene sets using the generally applicable gene-set enrichment for pathway analysis (GAGE) algorithm [106]. RNA sequencing data used for the gene expression analysis of both *atp6v1e1b*-deficient zebrafish and WT controls have been deposited in the ArrayExpress database at EMBL-EBI under accession number E-MTAB-8824, and can be accessed at the following link: https://www.ebi.ac.uk/arrayexpress/experiments/E-MTAB-8824.

## Metabolomics

In order to assess the effect of *atp6v1e1b* reduction on their metabolome, 3 batches of ∼100 *atp6v1e1b*[hi577aTg/hi577aTg], *atp6v1e1b*[cmg78/cmg78] and their WT controls were harvested at 3 dpf. Larvae were homogenized with a pestle in $dH_2O$ and the larvae were frozen until processed. Samples were subjected to Ultra Performance Liquid Chromatography High Resolution Mass Spectrometry (UPLC-HRMS) at the VIB Metabolomics Core Ghent (VIB-MCG). 10 μL was injected on a Waters Acquity UHPLC device connected to a Vion HDMS Q-TOF mass spectrometer Chromatographic separation was carried out on an ACQUITY UPLC BEH C18 (50 × 2.1 mm, 1.7 μm) column from Waters, and temperature was maintained at 40˚C. A gradient of two buffers was used for separation: buffer A (99:1:0.1 water:acetonitrile:formic acid, pH 3) and buffer B (99:1:0.1 acetonitrile:water:formic acid, pH 3), as follows: 99% A for 0.1 min decreased to 50% A in 5 min, decreased to 30% from 5 to 7 minutes, and decreased to 0% from 7 to 10 minutes. The flow rate was set to 0. 5 mL min−1. Both positive and negative Electrospray Ionization (ESI) were applied to screen for a broad array of chemical classes of metabolites present in the samples. The LockSpray ion source was operated in positive/negative electrospray ionization mode under the following specific conditions: capillary voltage, 2.5 kV; reference capillary voltage, 2.5 kV; source temperature, 120˚C; desolvation gas temperature, 600˚C; desolvation gas flow, 1000 L h−1; and cone gas flow, 50 L h−1. The collision energy for full MS scan was set at 6 eV for low energy settings, for high energy settings (HDMSe) it was ramped from 28 to 70 eV. Mass range was set from 50 to 1000 Da, scan time was set at 0.1s. Nitrogen (greater than 99.5%) was employed as desolvation and cone gas. Leucine-enkephalin (250 pg μL−1 solubilized in water:acetonitrile 1:1 [v/v], with 0.1% formic acid) was used for the lock mass calibration, with scanning every 1 min at a scan time of 0.1 s. Profile data was recorded through Unifi Workstation v2.0 (Waters). Data processing was performed with Progenesis QI software version 2.4 (Waters) for chromatogram alignment and compound ion detection. The detection limit was set at maximum sensitivity. In ESI+ and ESI- ionization,

9795 and 8959 features were detected and aligned, respectively. Post-processing was done with the online MetaboAnalyst 4.0 tool (http://www.metaboanalyst.ca) [107,108]. Briefly, the features were normalized by log transformation and Pareto scaling upon statistical analysis. T-test with FDR<0.1 identified 329 features for *atp6v1e1b*[hi577aTg/hi577aTg] compared to WT controls and 521 features as significant for *atp6v1e1b*[cmg78/cmg78] compared to WT controls. Functional interpretation directly from m/z peaks was performed based on a well-established GSEA algorithm [108]. Identifications of significant m/z peaks (one-way ANOVA, FDR<0.02) was attempted with Chemspider including following libraries: Chebi, Humann Metabolome database, Golm Metabolome database, The national compound collection, Targetmol, Sigma Aldrich, Pubmed, Natural products Atlas, NIST Spectra, NIST, Mcule, Massbank, KEGG, FoodDB, Extrasynthese and Analyticon Discovery.

### N-Glycomics

In order to assess the effect of *atp6v1e1b* reduction on N-linked glycosylation, 3 batches of ∼400 *atp6v1e1b*[hi577aTg/hi577aTg], *atp6v1e1b*[cmg78/cmg78], and WT controls were harvested at 3 dpf. Yolks were manually removed from all samples and the larvae were frozen until processed. Larvae were homogenized with a pestle in 1% octyl-glucoside (Life Technologies) in $dH_2O$. Samples were concentrated by Amicon ultrafiltration (Merck Millipore, Burlington, MA, USA). Samples were subjected to trypsin and peptide-N-glycosidase treatment before solid-phase extraction on C18 cartridges. Samples were dried by vacuum rotation after which glycans were methylated. Methylated glycans were applied onto a MALDI stainless steel target (Bruker Daltonics, MA, USA) together with an equal volume of matrix (α-cyano-4-hydroxy-cinnamic acid, saturated solution in 50% acetonitrile/0.1% aqueous TFA). MALDI-MS analysis was performed in the positive ion reflectron mode (HV acceleration 25 kV) on an Ultra-fleXtreme MALDI-TOF-TOF mass spectrometer (Bruker Daltonics) using the acquisition software FlexControl 3.3 and the data evaluation software FlexAnalysis 3.3 (Bruker Daltonics). The mass range selected for detection was from 500 to 5.000 Da. In general, 5.000 laser shots were accumulated at 1 kHz acquisition for each sample spot. Partial sequencing of glycans by MALDI-MS/MS was performed by laser-induced dissociation (LID) in the post-source decay (PSD) mode. Mass annotation was performed in FlexAnalysis and peaklists were generated after manual inspection of the entire mass range for correct isotope peak annotation. To assist oligosaccharide identification on the basis of sugar composition, the automatic tool for sugar mass increments was used. The molecular masses $M^+Na$, $M^+Na$-32, and $M^+Na$-54 were compared with standard mass lists, however a final identification was based on combined MS and MS/MS data. Fragmentation patterns were evaluated manually using fragmentation ion masses predicted by the GlycoWorkbench software tool.

### Quantitative analysis and statistics

Data processing and statistical analyses were performed using Graph Pad Prism version 9 (GraphPad Software, San Diego, CA, USA), which was used to generate each of the graphs shown in the figures and to perform statistical tests * P-value < 0.05; ** P-value < 0.01; *** P-value < 0.001; **** P-value < 0.0001. One-way ANOVA with correction for multiple comparison using the Bonferroni method was used for the majority of the analysis, except if mentioned otherwise in the figure legend or Materials and s.

## Supporting information

**S1 Fig. Whole mount in situ hybridization (WISH) for *atp6v1e1* ohnologs in zebrafish embryos.** (A-C) Representative images demonstrating WISH for *atp6v1e1a* and *atp6v1e1b* at

different developmental time points (24 hpf (n = 7), 48 hpf (n = 7), 72 hpf (n = 7), 120 hpf (n = 7) and 168 hpf (n = 7)). Expression patterns of *atp6v1e1a* and *atp6v1e1b* are similar. Scale bar = 200 μm. (D) Schematic representation of the *atp6v1e1a* gene (transcript ENSDART00000008986.6). The orange line represents the position of the target site of the sgRNA used for CRISPR-Cas9-induced indel mutagenesis (*atp6v1e1a*[-/-]). (E) RT-qPCR analysis showed a significant decrease in expression of *atp6v1e1a* at 3 dpf in *atp6v1e1a*[-/-]. Gene expression levels of *atp6v1e1b* remained normal. Data are expressed as mean ± SD from 5 biological replicates. 2-way ANOVA with Tukey test for multiple comparison. (F-J) Whole-mount bone staining of 12-month-old adult *atp6v1e1a*[-/-] and their respective wild-type (WT) controls in dorsal (I, J) and lateral positions (F, G). N = 8 per genotype. Scale bar = 2 mm (A, B, C, D).
(TIF)

**S2 Fig. Knock-out of *atp6v1e1b* in zebrafish shows an early phenotypic read-out.** (A) Schematic representation of the *atp6v1e1b* gene (transcript ENSDAR00000050221.5). The orange line represents the position of the proviral insert (*atp6v1e1b*[hi577aTg/+]) or the target site of the sgRNA used for CRISPR-Cas9-induced indel mutagenesis (*atp6v1e1b*[cmg78/+]). (B) RT-qPCR analysis showed a significant decrease in expression of *atp6v1e1b* in both zebrafish models at 3 dpf. Data are expressed as mean ± SD from 5 biological replicates. (C) Band intensities of fluorescence signals of atp6v1e1 and α-tubulin were quantified and normalized to the corresponding WT controls. Data are expressed as mean ± SD from 3 biological replicates, in which 20 zebrafish larvae were pooled per sample. 2-way ANOVA with Tukey test for multiple comparison was used for statistical analysis. Different protein extracts were loaded on the immunoblot. (D-E) Immunoblot of lysates obtained from *atp6v1e1b*[hi577aTg/hi577aTg], *atp6v1e1b*[cmg78/cmg78], and their WT controls at 4 dpf. We confirmed equal loading by staining for α-tubulin. (F) Loss of *atp6v1e1b* causes embryonic mortality in zebrafish. Kaplan-Meier curves for survival of *atp6v1e1b*-deficient zebrafish. (G) Hatching pattern of *atp6v1e1b*-deficient zebrafish. *Atp6-v1e1b*-defcient zebrafish showed delayed or no spontaneous hatching (H) Touch-Evoked Escape test (TEE-test) of *atp6v1e1b*[hi577aTg/hi577aTg] larvae (n = 22) and their respective WT controls (n = 8), *atp6v1e1b*[cmg78/cmg78] (n = 26) larvae and their respective WT controls (n = 26). The reaction after touch with a thick blunt needle was scored as follows: (1) no movement, (2) local muscle contractions of the embryo, (3) short distance swim movement and (4) normal swim movement towards the edge of the Petri dish.
(TIF)

**S3 Fig. *Atp6v1e1b*-deficiency impairs protein level of endo(lyso)somal markers in zebrafish larvae.** (A) Immunoblot of lysates from whole *atp6v1e1b*[hi577aTg/hi577aTg], *atp6v1e1b*[cmg78/cmg78], and corresponding WT control larvae at 4 dpf. We confirmed equal loading by staining for α-tubulin. (C-F) Quantification of the relative protein levels of EEA1 (marker for early endosomes), Rab5 (marker for early endosomes), Rab7 (marker for late endosomes), and LAMP1 (marker for lysosomes), normalized to total α-tubulin levels. Data are expressed as mean ± SD from 4 biological replicates. (G) Schematic presentation of the area covered by confocal microscopy in I. (H) The number of acidified vesicles is significantly reduced in *atp6v1e1b*[hi577aTg/hi577aTg], *atp6-v1e1b*[cmg78/cmg78], and Baf A1 treated WT larvae at 3 dpf compared to WT controls. (I) Representative confocal images of the cranial area are shown. Outlines of the eyes are indicated by dotted lines. Acidified vesicles in brain cells are visualized in red, including lysosomes, endosomes and exosomes. *Atp6v1e1b*[+/+] (n = 12), *atp6v1e1b*[hi577aTg/hi577aTg] (n = 12), *atp6v1e1b*[cmg78/cmg78] (n = 12) and *atp6v1e1b*[+/+] treated with 1.6 μM Baf A1 (n = 9). Scale bar = 100 μm.
(TIF)

**S4 Fig. Tissue specific expression and localization of early and late endosomal markers in *atp6v1e1b*-deficient zebrafish.** Representative images are shown. Confocal microscopy images of longitudinal sections of the yolk (A-D) and brain (E-H) from 4 dpf *atp6v1e1b*[+/+], *atp6v1e1b*[hi577aTg/hi577aTg], and *atp6v1e1b*[cmg78/cmg78] larvae stained for RAB5. (I-L) Confocal microscopy images of longitudinal sections of the muscle from the trunk of the zebrafish larvae from 3 dpf *atp6v1e1b*[+/+], *atp6v1e1b*[hi577aTg/hi577aTg], and *atp6v1e1b*[cmg78/cmg78] larvae stained for RAB7. Confocal microscopy images of longitudinal sections of the yolk (M-P) and brain (Q-T) from 3 dpf *atp6v1e1b*[+/+], *atp6v1e1b*[hi577aTg/hi577aTg], and *atp6v1e1b*[cmg78/cmg78] larvae stained for LAMP1. Negative control was imaged for each corresponding region in the zebrafish larvae. Scale bar = 20 μm.
(TIF)

**S5 Fig. *In vivo* loss of *atp6v1e1b* does not affect iron availability and does not induce a hypoxia-mediated response.** (A-H) No difference in gene expression of the HIF-1α-target genes, *egln3*, *vegfaa* and *pfkfb3*, were found in *atp6v1e1b*-deficient zebrafish. Interestingly, the gene expression of the HIF-1α-target genes, *vegfab*, *slc2a1a*, *slc2a1b*, *angptl4* and *pdk1*, was downregulated to varying extents in *atp6v1e1b*-deficient zebrafish based on RT-qPCR. Data are expressed as mean ± SD from 5 biological replicates. (I) Supplementation of 100 μM iron (Fe$^{3+}$) ammonium citrate (FAC) does not improve survival of *atp6v1e1b*-deficient zebrafish. Kaplan-Meier curves for survival of *atp6v1e1b*-deficient zebrafish with and without FAC treatment. (J) A*tp6v1e1b*-deficient and WT control zebrafish at 3 dpf exhibited normal levels of hemoglobin upon o-dianisidine staining. *Atp6v1e1b*[hi577aTg/hi577aTg] (n = 15) and their respective WT controls (n = 15), *atp6v1e1b*[cmg78/cmg78] (n = 15) and their respective WT controls (n = 15). Scale bar: 200 μM.
(TIF)

**S6 Fig. Reactome pathway analysis confirms enrichment of oxygen exchange, ion delivery and inflammatory signaling clusters in *atp6v1e1*-deficient zebrafish.** (A) RT-qPCR analysis of 10 from the top150 DEGs validating the results obtained in the unbiased transcriptomic analysis. Data are expressed as mean ± SD from 5 biological replicates. (B-C) Cluster plot showing the top enriched reactome pathways of the DEG from whole-body samples of *atp6v1e1b*-deficient and WT control zebrafish at 3 dpf.
(TIF)

**S7 Fig. Minor changes in phospholipid and fatty acid levels upon *atp6v1e1b* depletion in zebrafish larvae.** (A-H) HILIC LC MS/MS lipidomic analysis demonstrates minor changes in the total levels of phospholipids and fatty acids in 3 dpf *atp6v1e1b*-deficient zebrafish. Data are expressed as mean ± SD from 3 biological replicates. TG: triacylglycerides; DG: diacylglycerides; Chol: cholesterol; CE: cholesterol esters; PE: phosphatidylethanolamine; PE-O: 1-alkyl,2-acylphosphatidylethanolamines; PE-P: 1-alkenyl,2-acylphosphatidylethanolamines; LPE: lyso-phosphatidylethanolamine.
(TIF)

**S8 Fig. Modulating the ceramide synthesis pathway or the Krebs cycle does not ameliorate the survival of *atp6v1e1b*-deficient zebrafish.** (A-I) Survival curves are shown for *atp6v1e1b*-deficient zebrafish and controls until 8 dpf. Compounds were administered at 1 dpf, after chorion removal. (A-C) Administration of compounds interfering with the ceramide biosynthesis and cholesterol esterification. (D-I) Administration of compounds influencing the Krebs cycle. Kaplan-Meier curves for survival of *atp6v1e1b*-deficient and treated *atp6v1e1b*-deficient zebrafish.
(TIF)

**S9 Fig. Acute pharmacological inhibition of v-ATPase function recapitulates the effects of** ***atp6v1e1b*-deficiency on mitochondrial oxygen consumption.** OCR response to oligomycin and rotenone/antimycin a in 2 dpf *atp6v1e1b*$^{hi577aTg/hi577aTg}$ (A-B) and *atp6v1e1b*$^{cmg78/cmg78}$ zebrafish (C-D) and their respective controls after administration of vehicle (A, C) or the v-ATPase inhibitor Baf A1 (B, D) with oligomycin and rotenone/antimycin a or after administration. (C-D) OCR in WT controls and *atp6v1e1b*$^{cmg78/cmg78}$ zebrafish after administration of vehicle in C or Baf A1 in D with oligomycin and rotenone/antimycin a or after administration. Data are expressed as mean ± SEM from 5 biological replicates in which 4 zebrafish were pooled. Baf A1: Bafilomycin A1; OCR: oxygen consumption rate.
(TIF)

**S1 Table. List of DEGs with unknown gene names.**
(DOCX)

**S2 Table. Pathway activity profile of metabolites from *atp6v1e1b*-deficient zebrafish.** (Left) GSEA algorithm which considers the overall ranks of features without using a significance cut-off. (Right) Mummichog algorithm which implements an over-representation analysis method that is not able to detect subtle changes. *The mummichog compound hits represent the number of significant compounds divided by the total number of compounds per pathway.
(DOCX)

**S3 Table. Zebrafish mutation and primers used for genotyping.**
(DOCX)

**S4 Table. Compound screen in zebrafish.**
(DOCX)

**S5 Table. Primer sequences for qPCR analysis.**
(DOCX)

**S1 Text. Supporting information.**
(DOCX)

## Acknowledgments

We would like to thank Ms. Petra Vermassen, Ms. Hanna De Saffel, Ms. Zoë Malfait, and Ms. Myriam Claeys for their technical support. Electron microscopy was performed in the TEM facility of the Nematology Research Unit. Secondly, we would also like to thank Ms. Jelena Jovanovic for the practical assistance with the seahorse experiments. Lipidomics was performed by Lipometrix, core facility at the KU Leuven (www.lipometrix.be). The Ghent University Hospital is member of the European Reference Network (ERN)-Skin, VascERN, and ReCONNET.

## Author Contributions

**Conceptualization:** Lore Pottie, Patrick Sips, Bert Callewaert.

**Data curation:** Lore Pottie, Wouter Van Gool.

**Formal analysis:** Lore Pottie, Wouter Van Gool.

**Funding acquisition:** Paul Coucke, Bert Callewaert.

**Investigation:** Lore Pottie, Michiel Vanhooydonck, Franz-Georg Hanisch, Geert Goeminne.

**Methodology:** Lore Pottie.

**Project administration:** Lore Pottie, Bert Callewaert.

**Resources:** Franz-Georg Hanisch, Geert Goeminne, Andreja Rajkovic, Paul Coucke, Bert Callewaert.

**Software:** Wouter Van Gool.

**Supervision:** Patrick Sips, Bert Callewaert.

**Validation:** Lore Pottie.

**Visualization:** Lore Pottie.

**Writing – original draft:** Lore Pottie.

**Writing – review & editing:** Lore Pottie, Wouter Van Gool, Michiel Vanhooydonck, Franz-Georg Hanisch, Geert Goeminne, Andreja Rajkovic, Paul Coucke, Patrick Sips, Bert Callewaert.

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
