## [Decision Letter · Decision Letter 0]

5 Dec 2020

Dear Dr Callewaert,

Thank you very much for submitting your Research Article entitled 'Loss of a subunit of vacuolar ATPase identifies unexpected biological signatures of reduced organelle acidification in vivo' to PLOS Genetics.

The manuscript was fully evaluated at the editorial level and by 4 independent peer reviewers. The reviewers appreciated the attention to an important problem, but raised some substantial concerns about the current manuscript. Based on the reviews, we will not be able to accept this version of the manuscript, but we would be willing to review a much-revised version. We cannot, of course, promise publication at that time.

If you decide to revise the manuscript for further consideration at PLOS Genetics, please aim to resubmit within the next 60 days, unless it will take extra time to address the concerns of the reviewers, in which case we would appreciate an expected resubmission date by email to plosgenetics@plos.org.

Accompanying reviewer attachments are included with this email; please notify the journal office if any appear to be missing. They will also be available for download from the link below. You can use this link to log into the system when you are ready to submit a revised version, having first consulted our Submission Checklist.

[LINK]

We look forward to seeing your revised manuscript. Please do not hesitate to contact us if you have any concerns or questions.

Yours sincerely,

Zsolt Urban, Ph.D.

Guest Editor

PLOS Genetics

Gregory Barsh

Editor-in-Chief

PLOS Genetics

Reviewer's Responses to Questions

**Comments to the Authors:**

**Reviewer #1**

Please find enclosed the comments to the research article “Loss of a subunit of vacuolar ATPase identifies unexpected biological signatures of reduced organelle acidification in vivo, PGENETICS-D-20-01563” submitted for publication in PLOS Genetics by Bert Callewaert and colleagues. The paper comprises a set of interesting data important for a better understanding of cutis laxa conditions from a pathophysiological perspective. Furthermore, these date show that interfering with v-ATPase function in vivo also changes several pathways not obviously connected to this protein complex.

Major points:

The title to me would benefit from mentioning the organism analyzed and the deficient gene.

Introduction.

This section provides the reader with the function of the v-ATPase in general, with a broad list of mendelian conditions due to pathogenic variants affecting v-ATPase subunits and many pathways connected to this protein complex function. Some of these information’s are not necessary for the present paper and should be removed. Additionally, some of the knowledge presented here, is not further used throughout the manuscript/discussion and some of the pathways mentioned and very likely related to the multisystem phenotype in cutis laxa (e.g. AMPK) were unfortunately not investigated in the present in vivo model. Additionally, no information on tissue specific expression/assembly of different v-ATPase components have been provided which could be important for the discussion of the results. I recommend a restructuration of the introduction to allow the reader a better understanding of the project performed and to provide a more focused basis for the data and their discussion.

Results

- In supplementary figure 1, the effect of the mRNA expression in both zebrafish lines have been analyzed. Additionally, both lines lead to a strong reduction of the mRNA. Have the authors tested the protein expression? I know about the problem that sometimes no working antibodies are available for zebrafish, however, ATP6V1E1 is highly conserved. Maybe one of the described antibodies work.

- Have the authors observed a less severe phenotype in the line with the insertion within the 5’UTR. For example the vascular phenotype seems to be less severe. There is still some mRNA left which, due to the fact that the ORF should be intact, might get translated into active protein. Can the authors comment on that and include some words into the discussion?

- A more intuitive presentation of the EM data showing the matrix alterations would be of help. Can the authors show all the pictures in the same orientation?

- Complex glycan structures: It is not clear why some of the lines investigated (3/4) do not show any presens of these structures. Is this only due to the scaling of the y axis or are these structures not present at all. If so, why is the glycosylation phenotype more pronounce in the line carrying the frameshift (see above, differences of the manifestations) Can the authors discuss this point also with respect to potential Golgi abnormalities (see below)?

- The point about the endosomal and lysosomal alterations is to strong and not conclusively supported by the data. The alteration on EEA1 is impressive, however, the changes on Rab5 and Lamp1 seem to be minor. Additionally, have the author investigated the localization/morphology of these vesicles. Immunostaining of these markers (especially EEA1 and Lamp1) should be possible. The lysotracker staining is impressive. However, it is not immediately clear which part of the larvae was imaged. Could the authors show the structure by dotted lines within the picture or is a quantification possible? I do agree about alteration within the endosomal and lysosomal system, which is somewhat expected, however, I suggest weakening the point a bit.

- The analysis of the HIF target genes and the rescue experiments are interesting. I understand the point that the analysis of the iron “uptake” is crucial. However, from my point of view this seems not be altered at all. This to me leads to some lack of clarity within the complete manuscript. Focusing on the strongly altered pathways identified would lead to more clarity.

- The transcriptome analysis provides a lot of relevant information into the mechanisms leading to CL. The authors state, that the main candidate genes mentioned in the introduction were only minorly changed. Have the authors validated some of the candidates (OXPHOS….) and these initially mentioned genes pointing into distinct pathways (mTOR, AMPK) by qPCR? Additionally, changes in these pathways are not necessarily detectable via transcriptomics. For Notch, mTOR and AMPK signaling, good antibody kits are available which could help to identify potential misregulations/activations of them. Did the authors perform comparative proteomics?

- The lipidomics data are interesting. As a non-expert in zebrafish biology I was wondering where developmentally the origin of the yolk is. Due to the fertilization process this is very likely produced within the fish egg. If so, are these finding rather due to altered egg synthesis/composition of the yolk provided by the heterozygous mother? Have the authors found similar changes in specific celltypes of the larvae itself?

- The OCR data presented are interesting. To me, and in comparison, to some literature the reaction to Rotenone/Antimycin seems relatively slow. Can the authors state on this?

- One cellular feature observed in ATP6V1E1 related CL was a dilated Golgi. Did the authors found any evidence for this in the EM data or using immunolabeling?

Discussion:

The discussion at many points recapitulates the findings presented within the results section. Additionally, the strong focus on the expected, but not found iron related phenotypes gives the paper a strong imbalance towards this point. Other features observed (e.g. hypotonia) were not discussed at all especial in relation to the mitochondrial alterations.

The discussion about the relation of v-ATPase and cancer is interesting, however, to me it is not clear why this point is so extensively discussed.

Finally, the data presented are interesting and relevant. I recomment to restructure the complete manuscript with a focus on the altered pathways identified. I suggest starting with the presentation of the phenotype, glycosylation abnormalities and continue right away with the transcriptomics. From there, the argumentation towards the mitochondrial alterations, for example, are clear. All the findings should be discussed in more detail in relation and in comparison to physiological processes and to the human disease.

Minor points:

Sometimes it is not immediately clear how many animals have been subjected to each analysis and how many technical replicates have been performed. This should be carefully checked throughout all datasets

In some datasets two WT lines have been analyzed whereas the second line is missing in some other sets of data. Why haven’t both control lines been used consequently?

In some cases the figure reference within the text doesn’t fit to the data presented in the figures (Example: In addition, downregulation of elnb expression (Figure 2A+2B+) occurred in both mutant zebrafish lines)

**Reviewer #2**

The authors describe their findings on two Atp6v1eb zebrafish models, one that they generated using CRIPR-Cas9, and one obtained from ZIRC. Both mutant lines show severe phenotype that include reduced survival rates, impaired touch-evoked escape response, maxillary and mandibular hypoplasia, vascular abnormalities, and some indication of cardiac abnormalities. Overall the phenotypic characterization of the two genetic mutants are useful for future therapeutic explorations. The ~omics profiling is excellent.

Major Comments:

1. The authors have shown increased total EEA1 and ~reduced Rab7 levels by western blot. These results, however, do not necessarily suggest impaired endosomal maturation, and could not support a strong interpretation that deficiency of v-ATPase plays a prominent role in the maturation of endo(lyso)somes, nor that they have shown mechanistically that in vivo genetic depletion of atp6v1e1b leads to .. reduced maturation of endosomal and lysosomal vesicles. They also refer to a paper that reviews vesicular acidification and v-ATPase, among other things, but no association with endolysosomal maturation. To support the conclusion of reduced endolysosomal maturation, the authors would need to show the levels membrane-bound endosomal markers as compared to cytosolic membrane markers. An alternative would be to show confocal images of these markers, or to show accumulation of cargos that go through the endosomal pathway.

2. Although a hypothesis that LOF of Atp6v1eb in zebrafish leads to reduced acidification is not unexpected, the authors only showed lysotracker staining. This result will be strengthened by measurement of pH in lysosomes, or endolysosomal compartments.

3. The authors showed that FAC treatment did not rescue the survival rates of the two zebrafish mutants; also their data seem to indicate that there is no induction of HIF. It isn’t clear to me which dataset the authors used to support their initial hypothesis in pursuing this pathway, other than the previous report on ATP6V0 subunit. Are the authors thinking that the entire V-ATPase complex is destabilized upon loss of Atp6v1eb in zebrafish, or do they have results to show this? Do they have data showing reduced iron levels in their zebrafish models?

4. As a follow up to the previous question, I do not follow the seemingly surprising finding that there is a lack of hypoxia-mediated response in vivo. In human patients, there is an increased in trisialotransferrin, but I do not see similar data in the zebrafish – is transferrin abnormalities also seen in mutant Atp6v1eb zebrafish? In addition, there is no data showing detection of intracellular iron amounts in zebrafish mutants, that could potentially lead to HIF-mediated response.

5. The authors wrote, “Our data suggest alternative routes, bypassing the transferrin receptor pathway, to deliver iron to mitochondria in vivo” – but there is no evidence to suggest this.

Minor comments:

1. Mitochondrial profiling

a. Upon treatment of oligomycin, the OCR levels in the mutants were increased. How do the authors explain these results, given the reduced baseline OCR levels (which could indicate low ATP production.

2. Glycosylation profile:

a. What’s the significance of high mannose species in the mutant zebrafish lines? Does this correlate to an abnormal Golgi function since these species are glycoforms that indicate pre-Golgi processing?

b. The data seem to indicate that changes in complex glycans are more than the change in high mannose species – is this the case?

c. How would the author explain the differences in glycosylation pattern between the two mutant zebrafish lines? And Why is there a difference in the terminal sialic acid in the two mutant lines

3. Lipidomics

a. How do the authors interpret the increased amounts of DCER, but not CER.

4. Staining/phenotyping

a. Some of the figures are difficult to appreciate. For example, I really do not see the dilated mitochondria. It is not clear if both Atp6v1eb mutants have similarly coiled basement membrane. It isn’t clear if the authors are referring to loss of collagen structures – because the images seem to show some disorganization of the bundles

b. The lysotracker staining isn’t clear. The authors would need to show what area of the fish is stained, and which cells are stained.

c. Alcian blue showed maxillary and mandibular hypoplasia in both mutant zebrafish - Alcian blue shows reduced staining pattern – does this indicate bone cell abnormalities?

d. Some of the data describing survival show datapoints up to 150 dpf. I thought the mutants only survived up to 8 dpf?

5. How would you explain the reduced amount of collagen or elastin transcripts? Were these findings also seen in the transcriptomics studies?

**Reviewer #3**

The manuscript “Loss of a subunit of vacuolar ATPase identifies unexpected biological signatures of reduced organelle acifidication in vivo” by Lore P. et al. describes the various phenotypes of zebrafish atp6v1e1b KO mutant. The authors showed lots of data (transcriptome and metabolome) and I think that the manuscript seems to be interesting as it identified the roles of vacuolar ATPase function, but to be quite descriptive. Although the authors expressed their results as “unexpected biological signatures” in the title, almost all results in this manuscript are the “expected biological signatures” from the previous results in other model systems and human diseases. I think that the most important point in this manuscript is connection between organelle acidification through the vATPase and mitochondrial respiration. Are these two phenomena (organelle acidification by vATPase and mitochondrial respiration) tightly linked each other or is the mitochondrial respiration just side effect? I think the direct connection between them in cellular & molecular levels (some mechanistic analyses) is needed for publication in PLOS Genetics.

A minor point

In this manuscript, the authors mentioned that gene expression profile (HIF1a, Notch, Wnt, and mTOR signaling) in zebrafish atp6v1e1b KO mutant is different from that in in vitro systems. But there are two atp6v1e1 genes, atp6v1e1a and atp6v1e1b in zebrafish. Thus, they should discuss about this point.

Reviewer #4: In this work, the authors use mutant zebrafish larvae to investigate the physiological and cellular impacts of loss of function of the Atp6v1e1b subunit of the V-ATPase complex that pumps protons against concentration gradients. Interestingly, atp6v1e1b-/- mutant zebrafish develop several phenotypes—including craniofacial defects, cardiovascular defects, epidermal/skin defects, and protein glycosylation defects—that are reminiscent of problems observed in human patients with cutis laxa caused by mutations in ATP6V1E. The characterization and quantification of these phenotypes in zebrafish larvae, using two distinct atp6v1e1b-/- mutant lines, is comprehensive and convincing. These results suggest atp6v1e1b-/- mutant zebrafish provide a useful new animal model to study mechanisms underlying cutis laxa. The authors go on to use omics approaches to assess global changes in transcription, metabolism, and lipids in atp6v1e1b-/- mutants. These results give a broad overview of changes that occur in the mutants, and suggest several pathways/processes that are altered. A pharmacological approach is used to test whether targeting some of these processes could ameliorate phenotypes, but upon observing no positive effects, the authors conclude that targeting a single pathway is not sufficient to improve defects induced by loss of Atp6v1e1b. Overall, this work contributes to our understanding of in vivo consequences of altered V-ATPase function, but limitations of the global approach (pointed out below) appear to have prevented the authors from using their omics datasets to identify mechanisms that underlie specific phenotypes seen in mutant zebrafish (and human patients). Specific points are outlined below.

Major points:

1) There are a few caveats to using zebrafish genetics that should be clarified. One is that there are two zebrafish genes that encode for the V1E subunit-- atp6v1e1a and atp6v1e1b. It is not clear what cell types express either of these genes, and whether expression is overlapping (which may suggest functional redundancy) or cell-type specific. Clearly there are significant phenotypes and lethality in atp6v1e1b-/- mutants, but the atp6v1e1a gene should be discussed.

2) A second caveat is that V-ATPase subunits are maternally supplied (mRNA and possibly protein). This maternal supply is likely the reason why mutant embryos can develop relatively normally for the first couple of days and survive up to 8 days. The gradual onset of phenotypes in mutants suggests a gradual loss of maternal supply. Thus, it is not clear how much of the maternal Atp6v1e1b protein supply is depleted at 3 dpf when the global omics studies were done. This should be discussed.

3) A limitation to this study is that whole larvae were used to assess changes transcription, metabolism, lipids, and glycosylation. While V-ATPase probably has global cellular ‘house keeping’ functions, it is also likely that cell-type specific functions regulate processes such as Fe availability, hypoxia response, and signaling (mTOR, Notch, Wnt, etc) that underlie specific phenotypes. The authors touch on this in the Discussion, but this point could be expanded. Future work using cell-type specific approaches may uncover differences in mutants that shed light on mechanisms underlying mutant phenotypes in a given cell type.

Minor points:

1) For the retroviral insertion allele, I would predict the insertion in the 5’ UTR would interfere with mRNA transcription, and not induce NMD as stated. This should be clarified.

2) In Sup Fig 1C, is this dpf or hpf? The text indicates mutants only survive 8 days.

3) Pericardial edema in Fig 1A should be pointed out with an arrow.

4) For most analyses, the two atp6v1e1b-/- mutants have identical phenotypes. There are two exceptions reported: blood flow (Fig 1G) and complex N-glycans (Fig 2G) were altered in one mutant but not the other. It would be useful to comment on differences between the mutants.

5) With regard to the possibility of specific spatial and temporal functions for atp6v1e1b (see major points 1-3 above), it should be clarified what cell type and what stage of development we are looking at in Sup Fig 2G.

**Have all data underlying the figures and results presented in the manuscript been provided?**

Reviewer #1: Yes

Reviewer #2: Yes

Reviewer #3: Yes

Reviewer #4: None

PLOS authors have the option to publish the peer review history of their article (what does this mean?). If published, this will include your full peer review and any attached files.

Reviewer #1: No

Reviewer #2: No

Reviewer #3: No

Reviewer #4: No

---

## [Decision Letter · Decision Letter 1]

28 Mar 2021

Dear Dr Callewaert,

Thank you very much for submitting your Research Article entitled 'Loss of zebrafish atp6v1e1b, encoding a subunit of vacuolar ATPase, recapitulates human ARCL type 2 syndrome and identifies multiple pathobiological signatures of reduced vesicular acidification' to PLOS Genetics.

The manuscript was fully evaluated at the editorial level and by 4 independent peer reviewers. The reviewers appreciated the attention to an important topic but identified some concerns that we ask you address in a revised manuscript

We therefore ask you to modify the manuscript according to the review recommendations. Your revisions should address the specific points made by each reviewer.

[LINK]

Yours sincerely,

Zsolt Urban, Ph.D.

Guest Editor

PLOS Genetics

Gregory Barsh

Editor-in-Chief

PLOS Genetics

We appreciate the improvements that the authors have made to address the reviewer's comments. A few minor revisions are required as described by the reviewers.

Reviewer's Responses to Questions

**Comments to the Authors:**

Reviewer #1: Please find enclosed the comments to the research article “Loss of zebrafish atp6v1e1b, encoding a subunit of vacuolar ATPase, recapitulates human ARCL type 2 syndrome and identifies multiple pathobiological signatures of reduced vesicular acidification, PGENETICS-D-20-01563_R1” submitted for publication in PLOS Genetics by Bert Callewaert and colleagues. This is article has been strongly revised by the authors and all comments from the reviewer were carefully addressed. I recomment publication with a minor revision.

Some minor points:

The authors use the term type 2 cutis laxa syndrome frequently. For example within the title and the abstract they state: “…recapitulates human ARCL type 2 syndrome “ and “…cause type 2 cutis laxa, a condition further characterized by…”. According to OMIM there are different forms of type 2 cutis laxa. I suggest within the title they should use the term ARCL type 2 C syndrome and within the abstract the plural; a group of conditions. Please carefully check the remaining manuscript for this point.

Manuscript with text changes:

Line 292: The authors state the golgi is more frequently seen within the WT. This to me is rather due to the sectional plane and due to the fact that the golgi is more fragmented within the mutants and therefore not frequently seen within the mutant sections. This should be clarified.

Figure S2: The atp6v1e1 protein quantification shows that in the fish with the proviral insert the protein amount is relatively strong in comparison to the CRISPAR mediated KO. In one lane from the later line a similar protein amount is detectable whereas the two other lanes show a complete absence of the protein, as expected. Can the authors state on this point?

Line 343: The authors state: “…To our surprise, atp6v1e1b-deficient zebrafish displayed similar total p70-S6K1 protein expression and phosphorylation levels as observed in WT controls…” To me it look like there might be changes, however they are minor. I think the interpretation might be taken because the two control lines are not congruent. I suggest clarification within the results section.

Reviewer #2: The authors have answered most of my questions. Thank you you very much for the Discussion points!

However, since the authors do recognize pH measurement isn’t straightforward in zebrafish, and could well be ascertained in future studies, I suggest softening the conclusion on acidification based on their current results, and at least mentioning the caveats on the Discussion.

Reviewer #3: The authors answered all my questions. I recommend that their manuscript should be published in Plos Genetics.

Reviewer #4: The authors have made substantial revisions that have largely addressed my previous comments and significantly improved the manuscript. However, this paper remains descriptive, and does not provide new mechanistic insight into any of the many phenotypes described. In addition, there are still parts of the paper that are confusing and require additional clarification.

Minor points:

1) In the first section of the Results (starting on line 136), the back-and-forth flow between Fig. S1 and Fig. S2 is confusing and difficult to follow. The text describes atp6v1e1b knockout first and then atp6v1e1a, whereas this is in the reverse order in the figures (Fig. S1= atp6v1e1a and Fig. S2= atp6v1e1b). I suggest making the description of these mutants easier to follow.

2) In Fig. S2D-E western blots, it is not described what is loaded in the three lanes for wt and mutants. It should be clarified is these are the same extracts or different extracts.

3) For Fig. S5I, is this dpf or hpf?

4) The data and conclusions regarding HIF-1 is confusing. The authors state (starting in line 290) “HIF-1�-target genes (egln3, vegfaa, 291 vegfab, slc2a1a, slc2a1b, pfkfb3, angptl4 and pdk1) were downregulated in atp6v1e1b-deficient zebrafish at 3 dpf (S5 Fig.A-H)” and then conclude “Taken together, we show that loss of atp6v1e1b does not alter the Notch and HIF-1� signaling pathways…” It should be clarified why the authors conclude that HIF-1� signaling is not affected.

5) In line 448 the authors conclude “While zebrafish deficient in atp6v1e1a survived normally to adulthood without any phenotype, suggesting that this gene is a non-functional ohnolog…” I suggest softening the statement that atp6v1e1a is ‘non-functional,’ since there may be unrecognized compensatory mechanisms that could be discovered in future work.

**Have all data underlying the figures and results presented in the manuscript been provided?**

Reviewer #1: Yes

Reviewer #2: Yes

Reviewer #3: Yes

Reviewer #4: Yes

PLOS authors have the option to publish the peer review history of their article (what does this mean?). If published, this will include your full peer review and any attached files.

Reviewer #1: No

Reviewer #2: No

Reviewer #3: No

Reviewer #4: No

---

## [Editor Report · Decision Letter 2]

17 May 2021

Dear Dr Callewaert,

We are pleased to inform you that your manuscript entitled "Loss of zebrafish atp6v1e1b, encoding a subunit of vacuolar ATPase, recapitulates human ARCL type 2C syndrome and identifies multiple pathobiological signatures." has been editorially accepted for publication in PLOS Genetics. Congratulations!

Yours sincerely,

Gregory Barsh

Editor-in-Chief

PLOS Genetics

Gregory Copenhaver

Editor-in-Chief

PLOS Genetics

Comments from the reviewers (if applicable):

**Data Deposition**

http://datadryad.org/submit?journalID=pgenetics&manu=PGENETICS-D-20-01563R2

**Press Queries**

---

## [Editor Report · Acceptance letter]

15 Jun 2021

PGENETICS-D-20-01563R2 

Loss of zebrafish atp6v1e1b, encoding a subunit of vacuolar ATPase, recapitulates human ARCL type 2C syndrome and identifies multiple pathobiological signatures. 

Dear Dr Callewaert, 

We are pleased to inform you that your manuscript entitled "Loss of zebrafish atp6v1e1b, encoding a subunit of vacuolar ATPase, recapitulates human ARCL type 2C syndrome and identifies multiple pathobiological signatures." has been formally accepted for publication in PLOS Genetics! Your manuscript is now with our production department and you will be notified of the publication date in due course.

With kind regards,

Katalin Szabo

PLOS Genetics

On behalf of:
